



# A low-cost and open-source approach for supraglacial debris thickness mapping using UAV-based infrared thermography

Jérôme Messmer[1,*] and Alexander R. Groos[1,2,*]

[1]Institute of Geography, University of Bern, 3012 Bern, Switzerland
[2]Institute of Geography, Friedrich-Alexander-Universität Erlangen-Nürnberg, 91508 Erlangen, Germany
[*]The authors contributed equally to this work.

**Correspondence:** Alexander R. Groos (alexander.groos@giub.unibe.ch)

**Abstract.** Debris-covered glaciers exist in many mountain ranges and play an important role in the regional water cycle. However, modelling the surface mass balance, runoff contribution and future evolution of debris-covered glaciers is fraught with uncertainty as accurate information on small-scale variations in debris thickness and sub-debris ice melt rates is only available for a few locations worldwide. Here we present a customised low-cost UAV for high-resolution thermal imaging of
mountain glaciers and a complete open-source pipeline that facilitates the generation of accurate surface temperature and debris thickness maps from radiometric images. First, a thermal orthophoto is computed from individual radiometric UAV images using structure-from-motion and multi-view-stereo techniques. User-specific calibration and correction procedures can then be applied to the raw thermal orthophoto to account for atmospheric and environmental influences that affect the radiometric measurement. The corrected thermal orhthophoto reflects spatial variations in surface temperature across the surveyed debris-
covered area. Finally, a high-resolution debris thickness map is derived from the corrected thermal orthophoto using in-situ measurements in conjuction with an empirical or inverse surface energy balance model that relates surface temperature to debris thickness. Our results from a small-scale experiment on the Kanderfirn in the Swiss Alps show that the surface temperature and thickness of a relatively thin debris layer (ca. 0-15 cm) can be mapped with high accuracy. On snow and ice surfaces, the mean deviation of the mapped surface temperature from the melting point ($\sim$0 °C) was 0.4 $\pm$1.0 °C. The root-mean-square
error of the modelled debris thickness was 1.2 cm. Through the detailed mapping, typical small-scale debris features and debris thickness patterns become visible, which are not spatially resolved by the thermal infrared sensors of current-generation satellites. The presented approach paves the way for glacier-wide high-resolution debris thickness mapping and opens up new opportunities for more accurate monitoring and modelling of debris-covered glaciers.

## 1 Introduction

Supraglacial debris is present in the ablation zone of many mountain glaciers worldwide (Scherler et al., 2018; Herreid and Pellicciotti, 2020) and can influence their mass balance, geometry and dynamics through the modification of sub-debris ice melt rates (e.g. Rowan et al., 2015; Ferguson and Vieli, 2020; Mayer and Licciulli, 2021; Rounce et al., 2021; Delaney and Anderson, 2022). Important sources of debris supply are steep headwalls, valley slopes and lateral moraines, and subglacial till (e.g. Anderson et al., 2018; van Woerkom et al., 2019). Debris can be mobilised and transported onto glaciers by gravitational



mass movements such as avalanches, landslides and rockfalls. Debris deposited on the glacier surface, as well as debris melted
out of the ice, accumulates in the ablation zone and often forms a continuous debris layer (e.g. Anderson et al., 2018; Wirbel
et al., 2018; Ferguson and Vieli, 2020; Mayer and Licciulli, 2021).

The thickness of supraglacial debris can range from less than 1 cm to more than 2 m and usually shows high spatial variability
across the ablation zone. The estimated global mean supraglacial debris thickness is in the order of 30-40 cm (Rounce et al.,
2021). However, debris thinner than 10 cm predominates, usually accounting for more than 50 % of the total debris-covered
glacier area (Rounce et al., 2021; McCarthy et al., 2022). While thin debris (less than a few centimetres) increases melting
due to its lower albedo compared to a clean-ice surface, thick debris insulates the underlying ice and reduces melting (Østrem,
1959; Nicholson and Benn, 2006; Evatt et al., 2015). Due to the reduced ice melt rate, mountain glaciers with an extensive
and relatively thick debris cover are typically larger and a have a flatter and lower-situated tongue than clean-ice glaciers in a
similar topographic and climatic setting (Scherler et al., 2011).

Several satellite remote sensing studies have found similar thinning rates at comparable elevations for debris-covered and
clean-ice glaciers, despite their geomorphological differences (e.g. Kääb et al., 2012; Gardelle et al., 2013). Enhanced melting
due to the presence of ice cliffs and supraglacial ponds in the debris-covered area and relatively low emergence velocities of
debris-covered glacier tongues are discussed as possible explanations for this peculiarity (e.g. Sakai et al., 1998; Steiner et al.,
2015; Brun et al., 2018; Miles et al., 2018; Anderson et al., 2021; Buri et al., 2021). However, testing these hypotheses and
determining the high spatial variability of ice melt rates across debris-covered areas remains a challenge, partly due to the lack
of high-resolution supraglacial debris thickness maps. For quantifying the impact of debris on the glacier mass balance and for
projecting the future evolution of debris-covered glaciers and their response to ongoing climate change, accurate information
on the spatial debris thickness distribution is urgently needed (Kraaijenbrink et al., 2017; Rounce et al., 2020, 2021).

Over the last two decades, various methods have been developed to determine the thickness of supraglacial debris. These
include in situ point measurements (e.g. Mihalcea et al., 2006), terrestrial tachymetric and photogrammetric measurements of
debris over ice cliffs (Nicholson and Benn, 2013; Nicholson and Mertes, 2017), and ground-penetrating radar measurements
along predefined transects (McCarthy et al., 2017). For glacier-wide mapping, debris thickness can be derived from remotely
sensed surface temperatures or surface elevation change rates using physical or empirical functions that relate surface temper-
ature or sub-debris ice melt to debris thickness (e.g. Mihalcea et al., 2008a, b; Foster et al., 2012; Juen et al., 2014; Rounce
and McKinney, 2014; Groos et al., 2017; Rounce et al., 2018, 2021). Debris thickness maps based on satellite remote sensing
data capture general debris thickness patterns in the ablation zone, but are not very accurate and cannot resolve the small-scale
debris thickness variability and the presence of supraglacial ice cliffs and ponds due to their relatively coarse spatial resolu-
tion. To better estimate the ablation and runoff contribution of debris-covered glaciers, high-resolution mapping techniques are
required.

Recent advances in terrestrial and unoccupied aerial vehicle (UAV) infrared thermography offer new possibilities for map-
ping glacier surface temperature and supraglacial debris thickness at centimetre resolution. Ground-based thermal infrared
(TIR) images have been used in previous studies to investigate thermal processes at the glacier surface (Aubry-Wake et al.,
2015, 2018) and to estimate supraglacial debris thickness (Herreid, 2021; Tarca and Guglielmin, 2022). However, with this ap-





proach it is difficult to survey larger areas and impractical to measure objects that are not in line of sight. UAVs equipped with light-weight TIR cameras have proven to be a suitable alternative for mapping surface temperatures on debris-covered glaciers in mountainous terrain (Kraaijenbrink et al., 2018). Two recent studies have successfully demonstrated that debris thickness can be simulated from radiometric thermal UAV imagery if they are carefully calibrated and corrected (Bisset et al., 2022; Gök et al., 2022). However, in the study by Bisset et al. (2022) the number of debris thickness measurements for validation

was small (n = 3) and in the study by Gök et al. (2022) the methodological uncertainties (an RMSE of 6-8 cm for a mean debris thickness of 9 cm) were relatively large, highlighting the importance of further exploring and developing this mapping approach.

    Here we present a customised UAV for high-resolution thermal imaging on glaciers and a complete open-source pipeline that enables the generation of surface temperature and debris thickness maps from raw thermal imagery acquired with UAVs.

Compared to previous studies that mainly rely on proprietary software (e.g. Kraaijenbrink et al., 2018; Bisset et al., 2022; Gök et al., 2022), our open-source approach supports the generation of accurate raw thermal orthophotos that can be further processed and corrected according to the users' needs. We use thermal imagery and in situ measurements of debris thickness and debris temperature from the Kanderfirn in the Swiss Alps (Fig. 1) to illustrate and discuss the limitations and potential of the methodology.

## 2   Study site

The Kanderfirn (46.47 °N, 7.78 °E), a southwest-facing valley glacier in the Bernese Alps, was chosen as a test site for the UAV-based thermal imaging and debris thickness mapping as it comprises both a debris-covered and debris-free area. The latter is helpful for assessing the accuracy of the thermal images (see Section 3.6.4). Furthermore, experience with the use of UAVs on this glacier already exists from previous campaigns (Groos et al., 2019, 2022a). The tongue of the Kanderfirn is situated

at ca. 2300 m a.s.l. The highest point is the Petersgrat with an elevation of ca. 3200 m a.s.l. The area of the Kanderfirn was 16.0 km$^2$ in 1850. It decreased to 13.8 km$^2$ in 1973 (Maisch et al., 2000; Paul, 2003) and to 12.2 km$^2$ in 2010. Today it is less than 12.0 km$^2$ (Fischer et al., 2018; Groos et al., 2019). An area of around 0.8 km$^2$ in the northwestern part of the Kanderfirn is covered by debris (Swiss Federal Office of Topography, 2021). The predominant part of the debris cover on the Kanderfirn originates from the south face of the Blüemlisalp, which mainly consists of sedimentary rocks. In the southwest, granites and

gneisses can be found. The lithology at the base is a mix of sedimentary and igneous rocks (Hügi, 1956 in Richter, 1974; Swiss Federal Office of Topography et al., 2005). Since the entire debris-covered area is too large to be mapped with the customised quadcopter (see Section 3.1), only part of it was surveyed in this study (see Fig. 1).





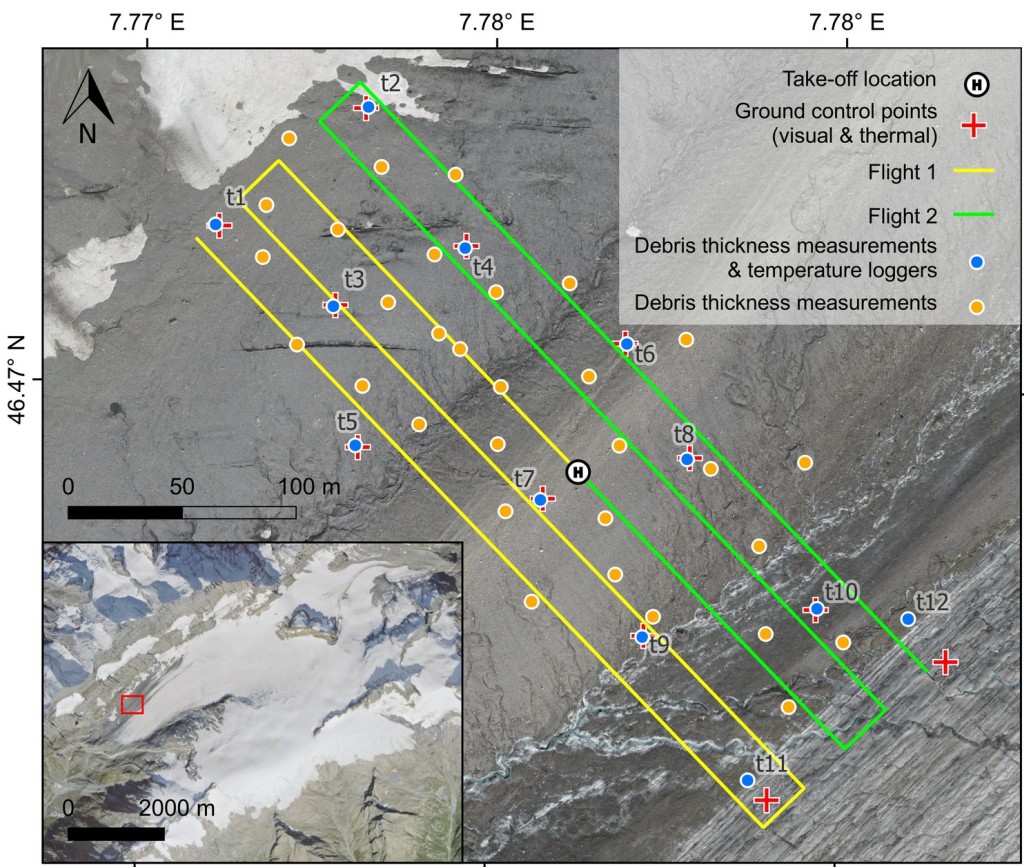

**Figure 1.** Overview map of the thermographic UAV surveys and in situ measurements on the Kanderfirn on 28 September 2021. The location of the UAV take-off site, ground control points, debris temperature loggers and debris thickness measurements is indicated by the respective symbols. Data source of the lower left map inset: SWISSIMAGE from 2018 (Swiss Federal Office of Topography, 2021).

## 3 Data and methods

### 3.1 Customised low-cost UAV

A small commercial rotary-wing UAV, a DJI Mavic Pro, equipped with a light-weight TIR camera (see Fig. 2) was chosen for the thermal imaging as the entire system can be easily transported to remote locations and ensures the safe take-off and landing of expensive and fragile sensors. The quadcopter weighs 750 g including its 43.6 Wh battery. According to the manufacturer, the maximum flight time in normal use (without additional payload) is 27 minutes. A global navigation satellite system (GNSS) and a forward and downward vision system is used for orientation. The quadcopter has a built-in stabilised and movable camera

with a focal length of 4.7 mm, equalling a 78.8° field of view. The maximum resolution for still images is 3000 × 4000 pixels. The UAV and its camera are controlled using a handheld remote controller and smartphone (DJI, 2016).



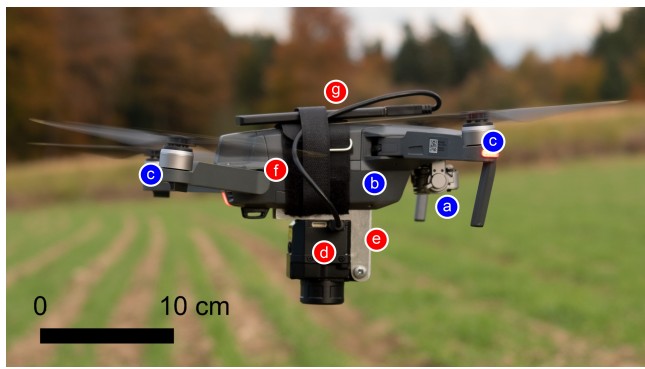

**Figure 2.** The complete UAV setup in action during a test flight. Parts that belong to the Mavic Pro itself are marked in blue, additional parts in red: **a)** UAV camera, **b)** UAV body, **c)** collapsible UAV arms including motors and propellers, **d)** TIR camera (FLIR VUE Pro R 640), **e)** customised mounting to attach the camera to the UAV body, **f)** Velcro straps for the attachment of the camera mounting and external battery, **g)** battery and cable to power the camera.

The TIR camera used in this study was a FLIR Vue Pro R 640 as it has been specifically designed for UAV-based applications and provides thermal imagery of similar accuracy as other, more expensive light-weight TIR cameras (Sagan et al., 2019). The Vue Pro R 640 measures 7.8 × 4.5 x 4.5 cm, weighs 140 g and has a focal length of 13 mm. An uncooled vanadium oxide microbolometer is used for radiometric measurements of thermal infrared radiation in the 7.5-13.5 µm spectrum. Both still and moving images can be captured with a viewing angle of 45°. The resolution of the thermal images is 640 × 512 pixels (Teledyne FLIR, 2019b). The exposure time of the camera is not specified, but estimated to be around 1/100 s (Teledyne FLIR, 2019a; USGS UAS, 2019). The claimed measurement accuracy is ±5 °C or 5 % of the reading. An external battery must be connected to the camera for power supply.

The TIR camera was attached to the UAV using a customised mounting system consisting of two angled aluminium plates (Fig. 2). The plates are inserted into grooves on the underside of the UAV and additionally attached to the body with two adjustable Velcro straps. This design allows for a flexible alignment and quick attachment and removal of the camera. The camera is powered with an external 11.4 Wh battery. The UAV plus camera and mounting weighs about 1 kg. It must be noted that the DJI Mavic Pro is not designed to carry loads. Additional payloads reduce the maximum flight duration and increase the risk of motor overheating and damage.

## 3.2 Visual and thermal infrared UAV surveys

The UAV-based thermal imaging on the Kanderfirn was carried out on 28 September 2021 in the early afternoon during slightly overcast conditions (Table 1). As a compromise between ascent time, ground sampling distance (GSD) and image detail, we chose a flight height of 100 m above ground level (a.g.l.). At this height, the estimated GSD of the thermal images is 13 cm and the image footprint is 83 × 67 m. The flight speed was set to 8.5 km/h. At this speed, assuming an exposure time of 1/100 s, motion blur is 2.4 cm (18 % of the GSD). For the photogrammetric processing and orthophoto generation (see Section 3.5),



a lateral image overlap of at least 60 % is recommended. To account for that, the distance between parallel flight tracks was set to 21 m, resulting in a lateral image overlap of 75 %. A maximum flight time of 13 minutes could be achieved with the customised UAV. This translated into individual flight tracks of about 900–1000 m.

**Table 1.** Key figures of the UAV surveys on 28 September 2021.

| Flight | Take-off time | Flight duration | Height [m a.g.l.] | Number of images | Surveyed area [ha] | GSD [cm] |
|---|---|---|---|---|---|---|
| 1/2 | 13:51 | 9 min 29 s | 100 | 409 (RGB), 828 (TIR) | 7.7 (RGB), 5.1 (TIR) | 3 (RGB), 13 (TIR) |
| 2/2 | 14:29 | 11 min 12 s | 100 | 403 (RGB), 831 (TIR) | 7.7 (RGB), 5.1 (TIR) | 3 (RGB), 13 (TIR) |

The third-party mobile application Litchi was used to fly the UAV in autonomous mode (VC Technology, 2020). A cardboard box with a cut-out hole matching the TIR camera on the underside of the UAV served as a base for manual take-off. As landing on the box was difficult, the UAV was grabbed in hover mode from a low height after completion of the survey. The open-source geographic information system QGIS (QGIS Development Team, 2020) and the Litchi mission hub (VC Technology, 2021) served for flight planning. A UAV-based high-resolution surface model of the Kanderfirn from 2021 (Groos et al.,

2022a) was uploaded to the mission hub to ensure a somewhat constant flight height above the uneven terrain. We performed two consecutive flights to survey part of the debris-covered and debris-free glacier area (see Fig. 1). Both visual and thermal images were taken with an interval of 1 s. The thermal images were stored as radiometric JPGs on the camera's SD card as only this file type contains radiometric data and allows for the accurate calculation of surface temperatures. Visual images were stored in JPG format on the UAV's SD card.

Twelve visual and twelve thermal ground control points (GCPs) were distributed across the survey area (Figs. 1 and 3) for accurate photogrammetric processing of the aerial images and for accurate georeferencing of the orthophotos. We used red Teflon sheets (A2 paper size) as visual GCPs and cardboard sheets (A3 paper size) wrapped in aluminium foil as thermal GCPs. Aluminium foil is a suitable material for thermal GCPs because it has a much lower emissivity than ice, snow or debris and therefore appears cooler on thermal images (e.g. Kraaijenbrink et al., 2018). The position of the centre of each GCP was

measured using a differential GNSS device (Trimble Geo 7x).

### 3.3    In situ measurements

### 3.3.1    Debris thickness

As a basis for the calibration and validation of the applied debris thickness models (see Section 3.7), we measured supraglacial debris thickness at 43 points that were randomly distributed across the survey area (Fig. 1). Since the debris thickness was

only a few centimetres at most locations, holes could be manually dug in the debris layer until the ice surface was reached. We measured the depths of the debris with a folding ruler and determined the exact position of each in situ measurement with the differential GNSS device.





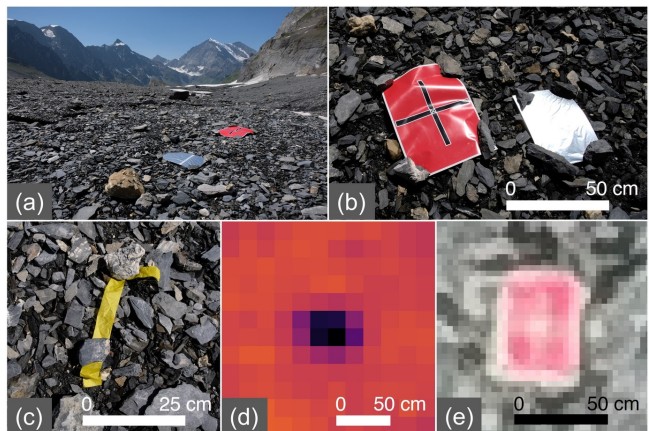

**Figure 3.** **a)** Overview of the surveyed debris cover (view towards the terminus), **b)** close-up view of a visual and a thermal GCP on the debris surface, **c)** yellow ribbon used to mark the position of the temperature loggers, **d)** a thermal GCP as seen in a thermal UAV image, **e)** a visual GCP as seen in a visual UAV image.

### 3.3.2  Debris temperature

At twelve locations, corresponding to the locations of the debris thickness measurements d1-d12, we installed button-sized,
splash-proof tempmate.B-2 temperature loggers in the debris (see Fig. 1) for comparison with the UAV-based thermal imaging (see Section 3.6). The loggers' measurement accuracy is ±0.5 °C in the range from -10 to 65 °C (Tempmate, 2022). Groos et al. (2022b) have shown in a previous study that the tiny temperature loggers are suitable for ground temperature measurements in high mountain environments. We placed each logger under a 0.5-1.0 cm thick shale stone to protect it from direct radiation and measured its position with the differential GNSS device. Because the temperature loggers are small and low in contrast to
the debris, we marked their location with yellow ribbon (Fig. 3). The loggers were set to record temperature at an interval of 10 minutes. The mean of all temperature readings between capture time of the first and last thermal image used for orthophoto creation was calculated for each temperature logger. Prior to the field campaign, we conducted an indoor comparison measurement at around 20 °C. In the comparison measurement, no noteworthy temperature deviations between the loggers were detected.

### 3.4  Meteorological data

We rely on meteorological data for the processing of the radiometric UAV images (see Section 3.6) and for the debris thickness modelling (see Section 3.7). Two automatic weather stations operated by the WSL Institute for Snow and Avalanche Research (SLF), one at Fisistock (46.4715° N, 7.6739° E, 2160 m a.s.l.) and one at Gandegg (46.4293° N, 7.7606° E, 2710 m a.s.l.), are located a few kilometers away from the Kanderfirn and continuously measure air temperature and relative humidity. We
calculated the mean air temperature and relative humidity for the period of the UAV surveys (13:00–15:00 h, CEST) and extrapolated the data from the two stations to the elevation of the study site (ca. 2460 m a.s.l.) using a simple linear regression





**Table 2.** Estimate of air temperature ($T_{air}$), relative humidity (RH), incoming shortwave radiation ($Q_S$) and incoming longwave radiation ($Q_L$) at the study site (ca. 2450 m a.s.l.) during the two UAV surveys (13:00–15:00 h, CEST) based on meteorological data from four weather stations in the vicinity of the glacier: Fisistock (2160 m a.s.l., $AWS_{low}$ for $T_{air}$ and RH), Gandegg (2710 m a.s.l., $AWS_{high}$ for $T_{air}$ and RH), Grächen (1600 m a.s.l., $AWS_{low}$ for $Q_S$ and $Q_L$) and Jungfraujoch (3570 m a.s.l., $AWS_{high}$ for $Q_S$ and $Q_L$). Data providers: Swiss Federal Office of Meteorology and Climatology (MeteoSwiss), WSL Institute for Snow and Avalanche Research (SLF).

| Parameter | Unit | $AWS_{low}$ | $AWS_{high}$ | Equation | Study site |
|---|---|---|---|---|---|
| $T_{air}$ | °C | 8.9 | 7.3 | $-0.0029 \cdot Ele_{Kanderfirn} + 15.2$ | 8.1 |
| RH | % | 83 | 61 | $-0.0391 \cdot Ele_{Kanderfirn} + 166.9$ | 71 |
| $Q_S$ | W m$^{-2}$ | 544 | 549 | $1/3 \cdot Q_{S\ Grächen} + 2/3 \cdot Q_{S\ Jungfraujoch}$ | 547 |
| $Q_L$ | W m$^{-2}$ | 320 | 262 | $1/3 \cdot Q_{L\ Grächen} + 2/3 \cdot Q_{L\ Jungfraujoch}$ | 281 |

model (Table 2). As the incoming shortwave and longwave radiation fluxes are not measured at these stations, we drew on data from two other weather stations, Grächen (46.1953° N, 7.8368° E, 1600 m a.s.l.) and Jungfraujoch (46.5476° N, 7.9854° E, 3570 m a.s.l.), which are located ca. 30 and 15 km away from the glacier and operated by the Swiss Federal Institute of Me-
teorology and Climatology (MeteoSwiss). We used the distance-weighted arithmetic mean to account for the varying distance between the glacier and both stations (Table 2).

## 3.5 Photogrammetric processing

We used structure-from-motion (SfM) and multi-view-stereo (MVS) techniques to generate a visual orthophoto, thermal orthophoto and digital surface model (DSM) from the UAV imagery. The visual orthophoto served as the basis for the ice, snow
and debris masks, and the thermal orthophoto for the surface temperature and debris thickness maps (see Sections 3.6 and 3.7). The DSM assisted the interpretation of the results.

We used only a subset of all acquired visual UAV images for the creation of the DSM and visual orthophoto to reduce the processing time. By selecting every eighth image, an overlap of 78 % could be achieved, guaranteeing still sufficient overlap in flight direction. The lateral image overlap, determined by the flight track distance (21 m), was about 85 %. All images taken
during flight direction changes were excluded from the selection. A total of 101 visual images from both UAV surveys were used for the subsequent photogrammetric processing.

The radiometric JPG files captured by the TIR camera display a colour gradient adjusted to the measured temperature range. The measured temperature values themselves are not retrievable outside of proprietary FLIR software. They are calculated by the camera from the sensor signal (raw data) and corrected by applying an undisclosed proprietary algorithm based on the
environmental parameters set prior to the capture. The colour gradient itself is useless for quantitative image analysis. To extract the raw data from the radiometric JPGs or to change the temperature correction settings after the acquisition, proprietary FLIR software packages are usually required (FLIR Systems, 2015; Teledyne FLIR, 2021b). This method has disadvantages, as these software packages are expensive and the algorithms used for temperature correction are not disclosed. We therefore drew on

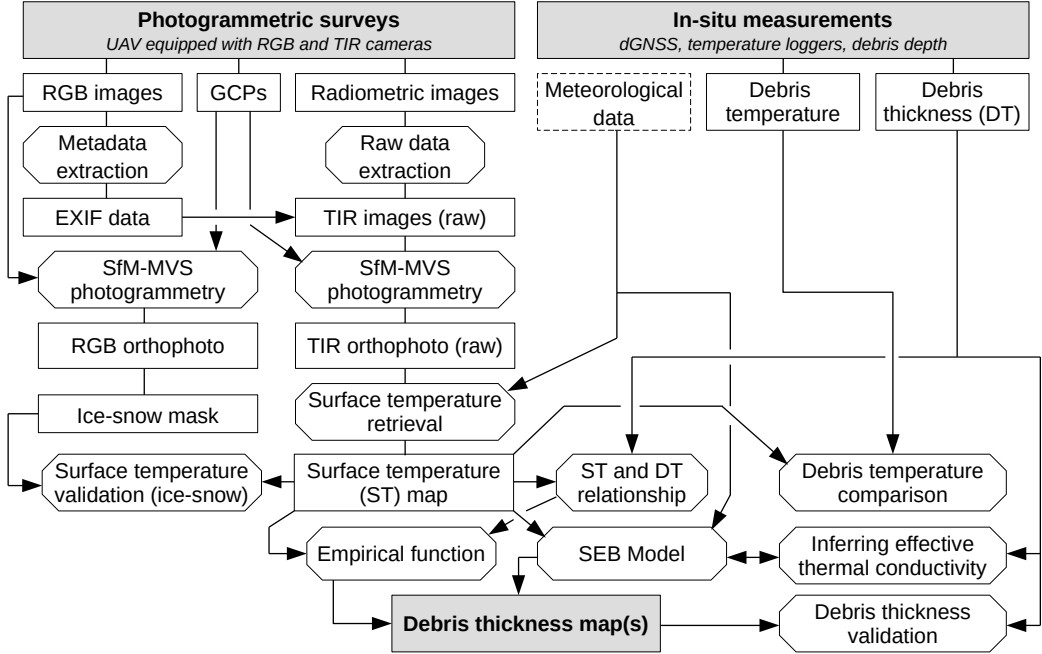

**Figure 4.** Open-source workflow for the processing of visual and radiometric UAV imagery from glacial environments and the generation of accurate high-resolution surface temperature and supraglacial debris thickness maps.

an open-source command-line application called ExifTool to retrieve the raw data from the radiometric images. ExifTool has

been developed for reading, writing and manipulating image, video and audio metadata (Harvey, 2021). The raw data contain dimensionless 16-bit number values corresponding to the electronic signal readout of the microbolometer sensor (Minkina and Dudzik, 2009; Pour et al., 2019; Tattersall, 2021b). Accurate surface temperatures could then be calculated from the raw values (see Section 3.6). As the radiometric images are not geotagged, positional information were copied from the metadata of the visual images with equivalent timestamps using the ExifTool and R (R Core Team, 2019; Harvey, 2021). A few blurred images

and those taken during changes in flight direction, take-offs and landings were excluded from further processing. By selecting every fifth image, a sufficient overlap of ca. 80 % in flight direction was achieved. The overlap between images from parallel flight tracks was around 75 %. A total of 162 raw thermal images from both UAV surveys were used for photogrammetric processing.

For the photogrammetric processing of UAV imagery (from glacial environments) and the generation of accurate orthophotos

and DSMs, both proprietary software (e.g. Kraaijenbrink et al., 2018; Bisset et al., 2022; Gök et al., 2022) and open-source software (e.g. Groos et al., 2019, 2022a) are available and have been successfully applied. We processed the selected visual and thermal images separately using the established proprietary software Pix4Dmapper (Pix4D, 2021a) to create a reference dataset for comparison with the output of our complete open-source pipeline (Fig. 4). From the visual images, we computed the DSM and visual orthophoto. Each of the twelve accurately measured visual GCPs (see Fig. 3 and Section 3.2) was referenced

in 5 UAV images and passed to Pix4Dmapper for georeferencing and accuracy optimisation. The spatial resolution was set to GSD and processing was done with optimal point density. Noise filtering and medium degree surface smoothing were applied for the DSM. From the thermal images, only a thermal orthophoto was computed. We passed 11 thermal GCPs, which were referenced again in 5 images, to Pix4Dmapper because one thermal GCP (no. 12) was not visible in enough images. Processing was done with high point density. The output resolution was set to GSD. By default, Pix4dmapper adjusts pixel values when

stitching images to create a visually pleasing image. With the reflectance map option, the adjustment of the pixel values can be skipped to display the pixel values faithfully (Pix4D, 2021b). Since the radiometric values were of central interest, this option was used.

As an open-source alternative for photogrammetric processing of the raw thermal UAV images, we tested OpenDroneMap (ODM, version 2.9.2, https://opendronemap.org/) with the application programming interface WebODM (version 2.2.0). We

did not process the visual UAV images with ODM because the general suitability of the software for the generation of accurate high-resolution orthophotos and DSMs of alpine terrain has been demonstrated before (see Groos et al., 2019, 2022a). We used the default settings in WebODM and only changed the point cloud density from medium to maximum (flag: pc-quality = ultra) to create a raw thermal orthophoto from the individual images. 11 thermal GCPs were passed to WebODM for georeferencing and accuracy optimisation.

### 3.6 Surface temperature retrieval

The raw thermal orthophotos computed with Pix4Dmapper and ODM contain dimensionless values corresponding to the electronic signal of the microbolometer sensor of the TIR camera. To obtain surface temperatures, the dimensionless values must be converted to the according radiation temperature incident on the thermal camera. Additionally, environmental influences that lead to discrepancies between the actual thermodynamic temperature of the surface and the radiation temperature, which

is measured by the thermal camera, must be corrected for (Minkina and Dudzik, 2009; Teledyne FLIR, 2016; Pour et al., 2019; Tattersall, 2019).

The conversion and correction was carried out with the raw2temp algorithm of the Thermimage R package (Tattersall, 2021a), which is based on the work of Minkina and Dudzik (2009). The output of the algorithm are corrected surface temperatures in °C. It has been shown that the raw2temp algorithm computes surface temperatures very similar to those of several

proprietary software packages (Tattersall, 2019), such as FLIR Tools (FLIR Systems, 2015) and FLIR Thermal Studio (Teledyne FLIR, 2021b). The algorithm firstly calculates an atmospheric transmission factor. Secondly, it calculates the portion of the signal attributable to the thermal radiation flux of the atmosphere and the thermal radiation flux reflecting off the investigated surface. Using the atmospheric transmission, emissivity and the interfering radiation fluxes, the surface temperature is calculated from the raw signal.

### 3.6.1 Atmospheric correction

To correct for the atmospheric influence on the radiometric measurements, information on the height of the air column between the glacier surface and sensor are important. The flight height was therefore taken into account. Moreover, knowledge of the





meteorological conditions at the study site during the subsequent UAV surveys are essential. As a rough estimate, we calculated the mean air temperature ($T_{\text{air}}$) and relative humidity ($RH$) at the study site during the UAV surveys using data from two nearby

weather stations and a simple linear regression model (see Section 3.4 and Table 2). Following the procedure of Tattersall (2021a), water vapour pressure ($P_{wv}$) was estimated from air temperature and relative humidity:

$$P_{wv} = (0.01 \cdot RH) \cdot exp(1.5587 + 0.06939 \cdot T_{\text{air}} - 0.00027816 \cdot T_{\text{air}}^2 + 0.00000068455 \cdot T_{\text{air}}^3). \tag{1}$$

It served for the determination of an atmospheric transmission correction factor ($\tau$)

$$\tau = ATX \cdot exp(-\sqrt{\frac{h_{UAV}}{2}} \cdot (ATA1 + ATB1 \cdot \sqrt{P_{wv}})) + (1 - ATX) \cdot exp(-\sqrt{\frac{h_{UAV}}{2}} \cdot (ATA2 + ATB2 \cdot \sqrt{P_{wv}})), \tag{2}$$

where $h_{UAV}$ is the flight height of the UAV. An overview of the different atmospheric attenuation constants is given in Table 3. The TIR radiance of the atmosphere ($raw_{\text{atm}}$) is calculated in raw units:

$$raw_{\text{atm}} = \frac{PR1}{PR2 \cdot (exp(\frac{PB}{T_{\text{air}}+273.15}) - PF)} - PO, \tag{3}$$

from which the fraction of the atmosphere's thermal radiation of the radiation flux arriving at the sensor ($raw_{\text{atm.attn}}$) is determined using the atmospheric correction factor ($\tau$) and the thermal emissivity ($\varepsilon$) of the surface material (debris or ice/snow):


$$raw_{\text{atm.attn}} = -\frac{(\tau^2 - 1) \cdot raw_{\text{atm}}}{\varepsilon \tau^2} \tag{4}$$

### 3.6.2  Reflected apparent temperature correction

The algorithm of Tattersall (2021a) further corrects for thermal radiation from the ambient air or surroundings objects that is reflected off the investigated surface. If surface reflectance is higher than 0 ($\varepsilon < 1$), the reflected thermal radiation influences

the surface temperature measurement (Cronholm, 2002; Minkina and Dudzik, 2009).

A suggested method to measure the reflected apparent temperature, which is the temperature ascribed to the radiation reflected by the investigated surface, is to use crinkled and reflattened aluminium foil. It is laid on the target surface as a highly reflective and diffuse reflector of ambient radiation. By measuring its radiation temperature with emissivity set to 1, where emissivity itself has no effect on the measurement, the reflected apparent temperature can be approximated (Cronholm, 2002).

The thermal GCPs met the criteria of such a diffuse reflector. Hence, the temperature of the centre of each thermal GCP was extracted from the thermal orthophoto calculated as outlined in this section but with emissivity set to 1. The mean of the temperature readings (-7.4 °C) of all eleven thermal GCPs visible in the thermal orthophoto was taken as the reflected apparent temperature for surface temperature retrieval. The reflected apparent temperature for each GCP is listed in Table 4.

Based on the reflected apparent temperature ($T_r$) and following Tattersall (2021a), the TIR radiance reflected by the surface

($raw_{\text{refl}}$) is calculated in raw units:

$$raw_{\text{refl}} = \frac{PR1}{PR2 \cdot (exp(\frac{PB}{T_{\text{r}}+273.15}) - PF)} - PO, \tag{5}$$





**Table 3.** Parameter values used for the calculation of surface temperatures from the raw TIR camera signal. The determination of relative humidity and air temperature is outlined in Section 3.4 and Table 2. Flight height is known from flight planning. Surface emissivity values were taken from the literature (Salisbury and D'Aria, 1992; Rivard et al., 1995; Kraaijenbrink et al., 2018; Mineo and Pappalardo, 2021) and all attenuation and calibration constants were read from the thermal image exif data.

| Parameter symbol and description | | Value | Unit |
|---|---|---|---|
| $h_{UAV}$ | Flight height of UAV | 100 | m |
| $\varepsilon_d$ | Thermal emissivity of debris | 0.95 | |
| $\varepsilon_i$ | Thermal emissivity of ice/snow | 0.97 | |
| $T_r$ | Reflected apparent temperature | -7.4 | °C |
| $T_{air}$ | Air temperature | 8.1 | °C |
| $RH$ | Relative humidity | 71 | % |
| $PR1$ | PlanckR1 camera calibration constant | 17096.453 | |
| $PR2$ | PlanckR2 camera calibration constant | 0.046412475 | |
| $PB$ | PlanckB camera calibration constant | 1428 | |
| $PF$ | PlanckF camera calibration constant | 1 | |
| $PO$ | PlanckO camera calibration constant | -215 | |
| $ATA1$ | Atmospheric attenuation constant A1 | 0.006569 | |
| $ATA2$ | Atmospheric attenuation constant A2 | 0.012620 | |
| $ATB1$ | Atmospheric attenuation constant B1 | -0.002276 | |
| $ATB2$ | Atmospheric attenuation constant B2 | -0.006670 | |
| $ATX$ | Atmospheric attenuation constant X | 1.9 | |

from which its fraction of the thermal radiation flux incident on the sensor ($raw_{\text{refl.attn}}$) is calculated:

$$raw_{\text{refl.attn}} = \frac{(1-\varepsilon)}{\varepsilon} raw_{\text{refl}} \qquad (6)$$

### 3.6.3 Emissivity correction and surface temperature determination

Lastly, the algorithm corrects for emissivity ($\varepsilon$). Emissivity is used to characterise materials by how effectively they emit energy in the form of thermal radiation. It is expressed as a value between 0 (perfect reflector) and 1 (perfect emitter/black body). Every real object has an emissivity lower than 1 and therefore emits less radiation than a hypothetical perfect emitter at the same temperature. Hence, if emissivity is not adjusted for, a TIR camera would measure a surface temperature lower than the actual surface temperature of the object (Avdelidis and Moropoulou, 2003; National Physical Laboratory, 2021; Teledyne 270 FLIR, 2021a). For this correction step, the assumed emissivity of the debris layer is needed (Pour et al., 2019; Tattersall, 2021b; Minkina and Dudzik, 2009).

The emissivity of a given surface depends on the material, surface properties, measurement temperature, measurement angle and measurement wavelength (Salisbury and D'Aria, 1992; Rivard et al., 1995; Mineo and Pappalardo, 2021; National Physical





**Table 4.** Reflected apparent temperature ($T_{ref.app.}$) derived from the radiation temperature ($\varepsilon = 1$) calculated for the centre of each thermal GCP (see Section 3.6.2).

| GCP | $T_{ref.app.}$ [°C] |
|---|---|
| tGCP1 | -4.6 |
| tGCP2 | -9.1 |
| tGCP3 | -8.2 |
| tGCP4 | -6.8 |
| tGCP5 | -8.7 |
| tGCP6 | -15.5 |
| tGCP7 | -1.8 |
| tGCP8 | -4.6 |
| tGCP9 | -1.0 |
| tGCP10 | -10.9 |
| tGCP11 | -10.1 |
| Mean | -7.4 |
| SD | ±4.0 |

Laboratory, 2021). Its estimate is therefore a source of uncertainty. As laboratory measurements were beyond the scope of this
study, we relied on values from the literature. The debris-covered area on the Kanderfirn is mainly made up of limestone and
marly shales. Locally, smaller amounts of ferrous limestone and igneous rock are present. The latter lithologies were neglected
for the emissivity estimate due to their scarcity.

Only limited literature is available on the emissivity of rock materials. Rivard et al. (1995) measured emissivities of around
0.95 in the wavelength of 7.5–13 μm for different limestone samples. Mineo and Pappalardo (2021) also measured emissivities
of around 0.95 for limestone and similar values for other calcareous rocks in the same wavelength. Hence, we assumed an
emissivity of 0.95 for the debris cover. However, the emissivity of the debris cover on the Kanderfirn may differ from this
estimate. For surface temperature calculation of the clean-ice and snow-covered areas, we used an emissivity of 0.97 (e.g.
Salisbury and D'Aria, 1992; Kraaijenbrink et al., 2018). Along with the atmospheric transmission and radiation fluxes described
above, the emissivity (either for debris or ice/snow) is used in the algorithm of Tattersall (2021a) to calculate a raw unit
equivalent to the actual surface temperature ($raw_s$) from the raw signal values ($raw$):

$$raw_s = \frac{raw}{\varepsilon\tau^2} - raw_{atm.attn} - raw_{refl.attn}, \tag{7}$$

from which the surface temperature ($T_s$) in °C is then determined by applying various calibration constants extracted from the
metadata of the TIR camera (see Table 3).

$$T_s = \frac{PB}{log(\frac{PR1}{PR2\cdot(raw.obj+PO)} + PF)} - 273.15 \tag{8}$$





### 3.6.4 Accuracy assessment of surface temperatures

Since the surface temperature of ice and snow on a summer day is expected to be close to the melting point (0 °C), the ice and snow areas surveyed with a UAV can help to assess the accuracy of the mapped surface temperatures (Kraaijenbrink et al., 2018). However, ice and snow surfaces that are partially covered with dark cryoconite are likely to have surface temperatures slightly exceeding 0 °C. We evaluated the deviation of the surface temperature from 0 °C over the debris-free and snow-covered areas of the test site on the Kanderfirn. Surface temperatures >5 °C were excluded from the comparison as they originated exclusively from rocks along the debris-margin and from those scattered over the snow and ice surfaces. However, only a small fraction of the validation area was affected (Pix4D: 3 %; ODM: 7 %).

To evaluate the calculated debris surface temperatures and detect possible biases, the means of the debris temperature logger readings during the two flights were compared to the corresponding temperature values in the corrected thermal orthophoto. For the position of each logger, the surface temperature was extracted from the corrected thermal orthophoto. The area mean of a 3x3-pixel window centred around the logger position was used to account for uncertainties in the thermal orthophoto and temperature logger position. However, not all potentially observed differences must be the result of measurement uncertainties, as the compared temperatures correspond to different definitions of surface temperature (Becker and Li, 1995; Kraaijenbrink et al., 2018). The temperature loggers record the thermodynamic temperature, which can be measured locally for a medium in thermal equilibrium. The radiation temperature captured by the thermal camera is only emitted from the surface skin layer of a depth in the order of a wavelength. Small-scale roughness and material variability (Minnis and Khaiyer, 2000), moisture and temperature gradients in the debris layer and at the debris-air interface further affect the temperature comparison (Kraaijenbrink et al., 2018).

### 3.7 Debris thickness mapping

A supraglacial debris thickness map can be derived from a corrected thermal orthophoto using either a local empirical relationship between debris thickness and surface temperature or an inverted glacier surface energy balance model (e.g. Mihalcea et al., 2008a, b; Foster et al., 2012; Juen et al., 2014; Rounce and McKinney, 2014; Groos et al., 2017; Rounce et al., 2018, 2021). We explored both approaches following the procedure below (see Fig. 4).

### 3.7.1 Empirical method

A site- and time-specific exponential function with two fitting parameters is suitable to predict debris thickness from surface temperature. The fit is iteratively compared to the measured data using different starting values for the fitting parameters, which are selected using non-linear least squares (e.g. Mihalcea et al., 2008a, b; Juen et al., 2014; Groos et al., 2017). To enable the point-wise comparison of measured debris thickness and mapped surface temperature, we extracted the surface temperature from the corrected thermal orthophoto at the location of each in situ debris thickness measurement using a 3x3-pixel matrix. The 43 measurement pairs were randomly split into a training and validation dataset. On the basis of the training dataset, an empirical best fit model was calculated using the debrisThicknessFit function from the glacierSMBM R package (Groos et al.,

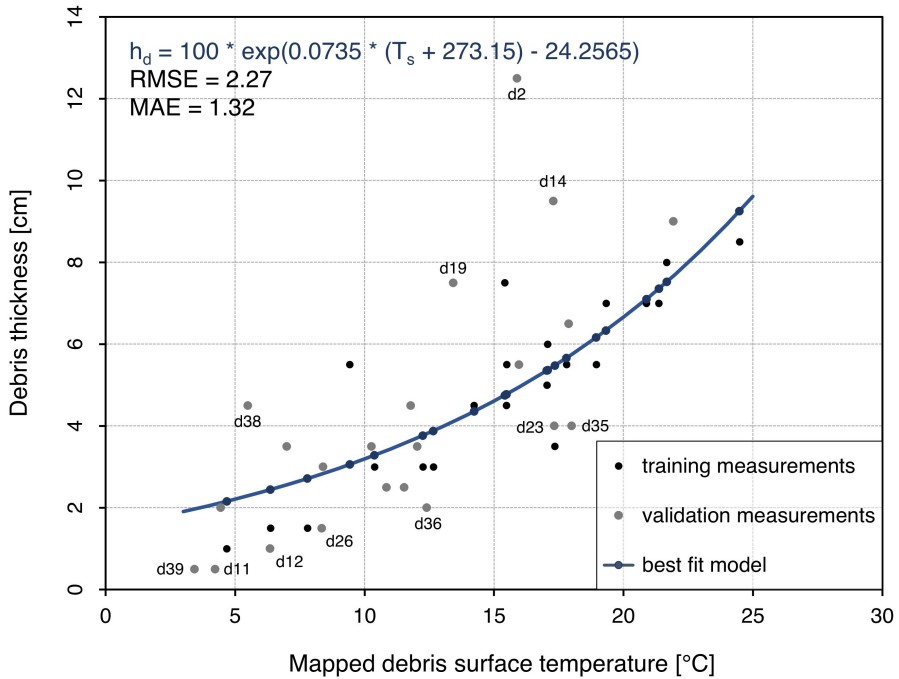

**Figure 5.** Empirical relationship between measured debris thickness and mapped debris surface temperature. The best fit model served for the extrapolation of the point data and the computation of a debris thickness map. Validation measurements that deviate considerably from the best fit model are labeled.

2017; Groos and Mayer, 2017):

$$h_d = exp(0.0735 \cdot T_s - 24.2565), \tag{9}$$

where $h_d$ is the predicted debris thickness (m) and $T_s$ is the debris surface temperature (K). The best fit model served for the
extrapolation of the point data and the computation of a debris thickness map using the debrisThicknessEmp function from the same R package. Half of the debris thickness measurements (n = 22) were considered for the validation. The uncertainty resulting from the empirical model was quantified using the root-mean-square deviation (RMSD = 2.3 cm) and mean absolute error (MAE = 1.3 cm).

### 3.7.2   Surface temperature inversion method

Surface energy balance models of varying complexity have been developed to predict ice melt beneath supraglacial debris using meteorological data and information on site-specific debris properties (e.g. Nicholson and Benn, 2006; Evatt et al., 2015). If solved for debris thickness, these models enable the calculation of debris thickness from surface temperature and the residual of the considered energy fluxes (e.g. Foster et al., 2012; Rounce and McKinney, 2014; Schauwecker et al., 2015; Groos et al., 2017; Rounce et al., 2021). We implemented and tested the theoretical sub-debris ice melt model of Evatt et al. (2015, Eq. 46)





as it has been proven to reproduce the characteristic features of the empirical Østrem curve (Østrem, 1959):

$$Q_M = \frac{\nu_1}{1 + \nu_2 h_{\mathrm{d}}} - \frac{\mu_1}{\mu_2 + e^{\gamma h_d}}, \tag{10}$$

where $Q_M$ is the sub-debris ice melt rate (m w.e. s$^{-1}$), $h_d$ is the debris thickness (m), $\gamma$ is a wind-speed attenuation constant
(234 m$^{-1}$) and the other terms are (Evatt et al., 2015, Eqs. 41-45)

$$\nu_1 = \frac{Q_L - \epsilon \sigma \overline{T}^3 + Q_S(1 - \alpha_d) + \beta T_{air}}{(1 - \phi)\rho_i L_m}, \tag{11}$$


$$\nu_2 = \frac{\beta + 4\epsilon \sigma \overline{T}^3}{k}, \tag{12}$$

$$\beta = \frac{\rho_a c_a u_*^2}{u_m - u_r(2 - e^{\gamma x_r})}, \tag{13}$$

$$\mu_1 = \frac{L_v u_*^2 (q_h - q_m) e^{-\gamma x_r}}{(1 - \phi)\rho_i L_m u_r}, \tag{14}$$

$$\mu_2 = \frac{(u_m - 2u_r) e^{-\gamma x_r}}{u_r}. \tag{15}$$

A summary of the units and values of the individual parameters and constants is provided in Table 5. Please refer to the original
publication for more information on the derivation of the individual equations. The surface temperature ($T_s$) of the debris layer
is

$$T_s = \frac{\rho_i L_m (1 - \phi)\nu_1 h_d}{k(1 + \nu_2 h_d)}. \tag{16}$$

After rearrangement, this function can be used to derive debris thickness from surface temperature and meteorological data:

$$h_d = \frac{kT_s}{\rho_i L_m \nu_1 - kT_s \nu_2}. \tag{17}$$

Since the thermal conductivity is a very sensitive parameter in sub-debris ice melt models, difficult to determine in the field
and, thus, usually unknown a priori (e.g. Steiner et al., 2021), we used this parameter for model calibration. Vice versa, this
allows the effective thermal conductivity of a debris layer to be estimated if its debris thickness and surface temperature are
known.





**Table 5.** Parameter values used for the surface energy balance model (Eq. 17). The first eleven parameter values are site-specific (see Sections 3.4 and 3.6.3). The albedo of the debris layer of the Kanderfirn was estimated using the glacier albedo map of Naegeli et al. (2019). The effective thermal conductivity was calibrated and the thermal emissivity of the debris layer was estimated (see Rivard et al., 1995; Mineo and Pappalardo, 2021). The last six parameters take non-site-specific values. Parameter values adopted from the original publication are indicated with an asterisk (see Evatt et al., 2015).

| Parameter symbol and description | | Value | Unit |
|---|---|---|---|
| $Q_S$ | Incoming shortwave radiation | 547 | W m$^{-2}$ |
| $Q_L$ | Incoming longwave radiation | 281 | W m$^{-2}$ |
| $T_{air}$ | Measured air temperature | 8.1 | °C |
| $q_m$ | Measured humidity level | 0.71 | $q_h$ |
| $q_h$ | Saturated humidity level | 0.008 | kg m$^3$ |
| $\rho_a$ | Air density | 0.95 | kg m$^3$ |
| $u_m$ | Wind speed* | 2.2 | m s$^{-1}$ |
| $x_m$ | Measurement height | 2.0 | m |
| $k$ | Thermal conductivity of debris | 1.0 | W m$^{-1}$ K$^{-1}$ |
| $\epsilon_d$ | Thermal emissivity of debris | 0.95 | |
| $\alpha_d$ | Debris albedo | 0.07 | |
| $x_r$ | Surface roughness height* | 0.001 | m |
| $u_*$ | Friction velocity* | 0.16 | m s$^{-1}$ |
| $u_r$ | Slip velocity* | $u_*$ | m s$^{-1}$ |
| $\phi$ | Volume fraction of debris in ice* | 0.01 | |
| $\gamma$ | Wind speed attenuation constant* | 234 | m$^{-1}$ |
| $c_a$ | Specific heat capacity of air | 1000 | J kg$^{-1}$ K$^{-1}$ |
| $L_m$ | Latent heat of melting ice | $3.34 \cdot 10^5$ | J kg$^{-1}$ |
| $L_v$ | Latent heat of water evaporation | $2.5 \cdot 10^6$ | J kg$^{-1}$ |
| $\overline{T}$ | Water freezing temperature | 273 | K |
| $\rho_i$ | Ice density | 900 | kg m$^3$ |
| $\sigma$ | Stefan-Boltzmann constant | $5.67 \cdot 10^8$ | W m$^{-2}$ K$^{-4}$ |

## 4 Results

### 4.1 Measured debris thickness and temperature

The supraglacial debris thickness measured manually at 43 points on the Kanderfirn (Fig. 1) ranges from less than 1 cm to about 13 cm (Table A1). Individual boulders scattered over the debris-covered glacier area are more than 50 cm thick, but they have not been measured and mapped explicitly. The average debris thickness is 4.6 cm and the median 4.5 cm. The distribution of point measurements has a positive skew, indicating that very thin debris thicknesses (<5 cm) predominate (Fig. 6). During the two UAV surveys on 28 September 2021, the mean debris temperature measured at 12 locations (Fig. 1) at a depth of ca.

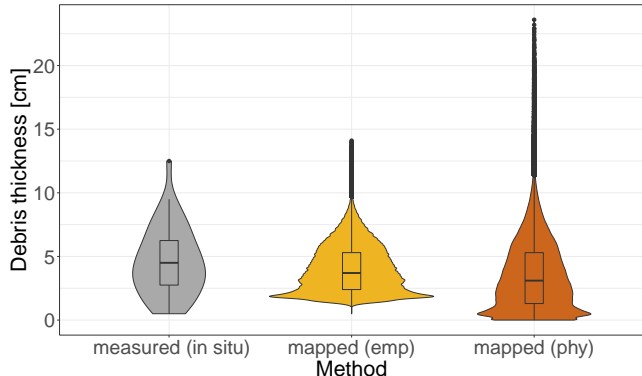

**Figure 6.** Distribution of the 43 in situ debris thickness measurements (see Fig. 1) and of all values from the empirical (emp) and physical (phy) debris thickness maps (see Figs. 12 and 14g).

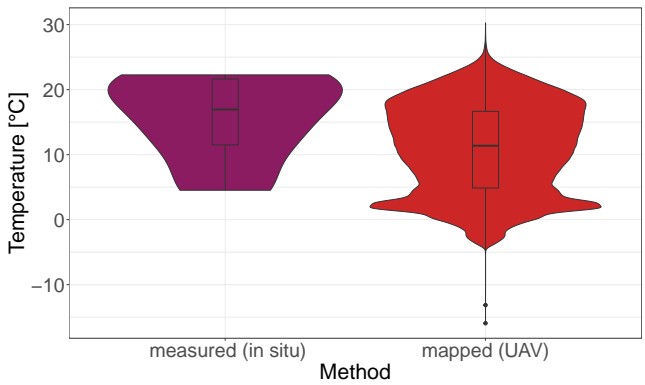

**Figure 7.** Purple (left): distribution of the 12 in situ debris temperature measurements at a depths of ca. 0.5-1.0 cm below the surface of the debris layer (Fig. 1). Red (right): all values from the surface temperature map (Fig. 10). Note that the low surface temperatures (outliers) are due to the thermal GCPs, as the very low emissivity of the aluminium foil was not taken into account when correcting the thermal orthophoto.



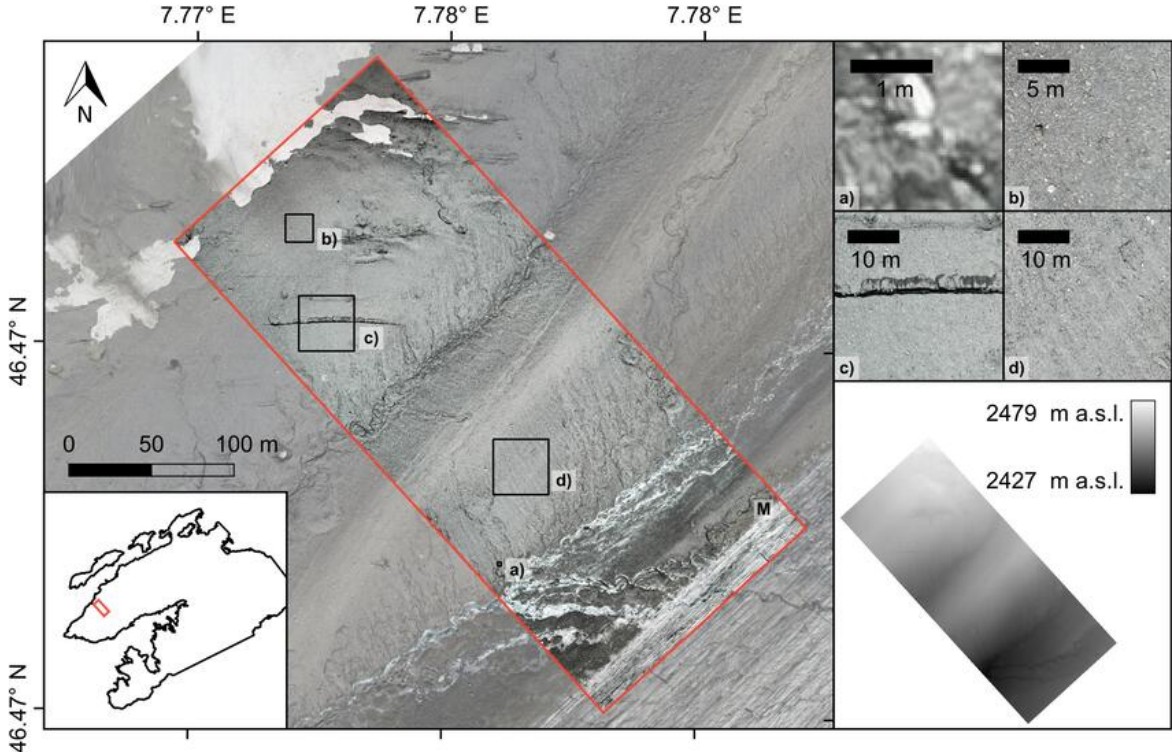

**Figure 8.** High-resolution visual orthophoto (see a) and DSM (map inset) of the glacier area surveyed on 28 September 2021, comprising clean ice, debris-covered ice (b, d), snow deposits from avalanches, crevasses (c), ice cliffs (c) and melt water channels. Background: UAV-based orthophoto from 14 September 2021.

0.5-1.0 cm varied spatially and ranged from 4.5 to 22.3 °C (Table A2). The mean debris temperature was 15.6 °C and the median 16.0 °C (Fig. 7).

### 4.2   Positional accuracy of visual and thermal orthophotos

The positional accuracy of the generated orthophotos is indicated in the quality report of the photogrammetry software. For the visual orthophoto, the accuracy obtained at 12 GCPs (Fig. 1) ranges from 3 to 18 cm (Table A3). The accuracy at most

of the GCPs is similar to or smaller than the positional uncertainty of the differential GNSS measurements itself. The root-mean-square error (RMSE) as a measure for the overall positional accuracy is 8 cm. For the thermal orthophoto, the accuracy obtained at 11 GCPs ranges from 5 to 22 cm (Table A4). The accuracy at the individual GCPs is similar to the positional uncertainty of the differential GNSS measurements. The RMSE of 9 cm is slightly higher than that of the visual orthophoto, but less than the GSD of 13 cm. The positional accuracy of the thermal orthophoto produced with ODM is similar to that of

Pix4D.





### 4.3 Surface characteristics and topography of the debris layer

The high-resolution orthophoto and DSM (GSD = 3 cm) cover a glacial area of ca. 6 ha and include clean ice, debris-covered ice, avalanche deposits, crevasses, ice cliffs and melt water channels (Fig. 8). Most of the surveyed area is covered with debris. Clean ice can be found in the southeastern part of the surveyed area and snow deposits from avalanches in the northwestern

part below the southface of the Blüemlisalp. One melt water channel runs more or less parallel to the debris-ice transition area and the other one forms a small valley, separating the debris-covered area in two elevated parts. The height difference between the lowest point of the melt water channel and the highest point of the debris-covered area is about 50 m. Ice cliffs occur close to the central stream and in both elevated debris-covered areas. Larger crevasses are restricted to the area northwest of the central stream. In the southeastern corner of the surveyed area, the lower tail of a supraglacial moraine (labeled with "M") that

has become wider over the last years can be seen (Fig. 8; for a better overview see Fig. 11 in Groos et al., 2019).

### 4.4 Difference between measured and mapped debris temperatures

Since the tiny temperature loggers recorded the debris temperature at shallow depths and the TIR camera mounted on the UAV measured the skin temperature of the debris layer, the in situ measurements and mapped surface temperatures cannot be expected to match exactly. However, a cross-comparison makes it possible to assess the plausibility of the mapped debris

surface temperatures. The cross-comparison shows that the measured debris temperatures and mapped surface temperatures

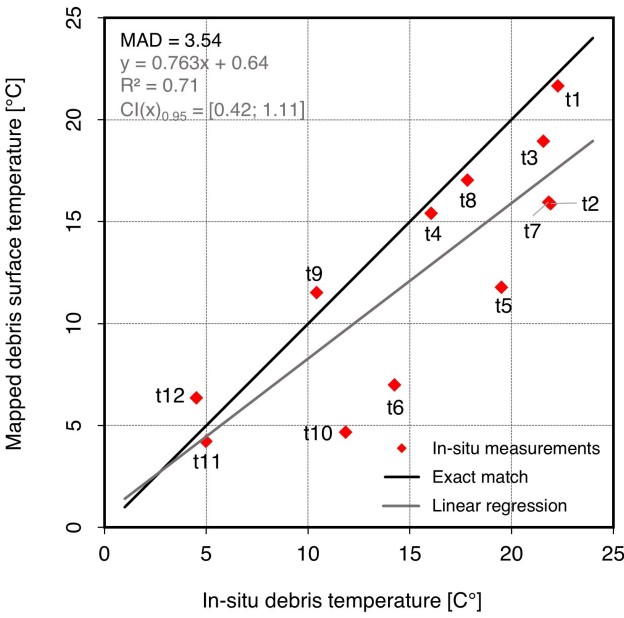

**Figure 9.** Debris temperature measured at 12 points on the glacier (Fig. 1) at a depth of ca. 0.5-1.0 cm below the surface of the debris layer versus mapped surface temperature extracted from the corrected thermal orthophoto (Fig. 10) for the same points.



agree relatively well (Fig. 9). The coefficient of determination ($R^2$) is 0.71. The difference between the measured and mapped temperatures ranges from -7.7 to 1.8 °C. At five points, the deviation exceeds 5 °C. However, the deviation at the other points is consistently less than 3 °C and in most cases less than 1 °C. The mean absolute difference is 3.5 °C.

### 4.5 Spatial variations in glacier surface temperature

The corrected thermal orthophoto that has been generated from the 162 raw thermal images has a GSD of 13 cm and covers an area of about 6 ha (Fig. 10). The difference between the surface temperature map derived from the raw thermal orthophoto generated with the open-source software (ODM) and the one derived from the raw thermal orthophoto generated with the established proprietary software (Pix4Dmapper) is negligible (0.2 ± 1.2 °C). Small-scale deviations originate from the imperfect co-registration of both maps and are not further discussed here. More interesting is the prominent large-scale, stripe-like pattern

tracing the flight tracks (Fig. B1), possibly caused by interference with solar radiation in the southeastern direction of flight. Why the signal is imprinted only in one of the two thermal orthophotos is not clear. Since the deviations are overall very small

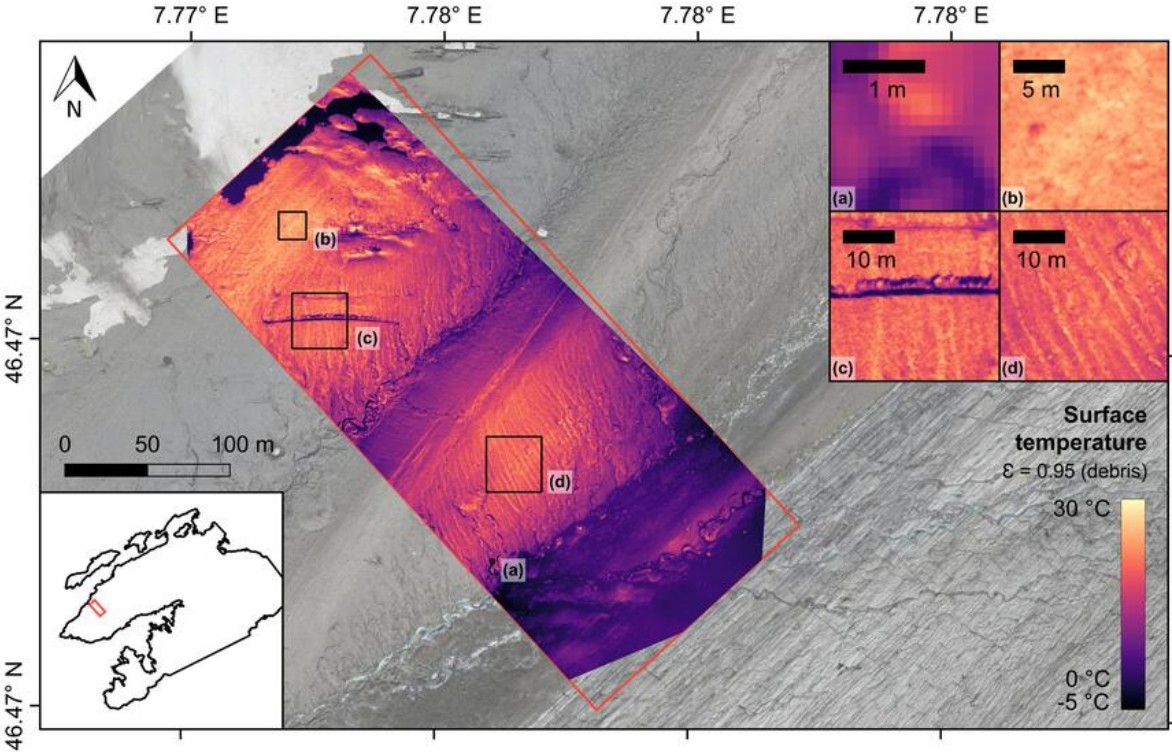

**Figure 10.** Corrected thermal orthophoto showing the surface temperature ($\epsilon = 0.95$) of the surveyed glacier area at about 2 p.m. on 28 September 2021 The four highlighted sections (a, b, c, d) are the same as in Fig. 8 and Fig. 12. Detail **a)** illustrates the spatial resolution, detail **b)** shows a debris-covered area with relatively high surface temperatures, detail **c)** includes a crevasse and an ice cliff and detail **d)** is an example of small-scale debris surface temperature variations. Background: UAV-based visual orthophoto from 14 September 2021.

and have little influence on the final debris thickness map, the spatial surface temperature variations are illustrated only for the thermal orthophoto that was obtained through the photogrammetric processing of the radiometric images with Pix4Dmapper. Spatially, the mapped surface temperatures vary between -4 and 30 °C, but most pixel values are between 0 and 25 °C (Fig. 7).

At the location of the thermal GCPs, surface temperatures down to -16 °C can be found. However, these values do not represent real surface temperatures, as the very low emissivity of the aluminium foil (<0.1) of the GCPs was not taken into account when processing the raw thermal orthophoto. During the period of the two UAV surveys, the glacier surface temperature averaged over the entire study area was about 11 °C (Fig. 7). Surface temperatures are lowest on the snow patches and highest in the elevated debris-covered area below the Blüemlisalp southface. Due to their much lower temperatures, small-scale features such

as crevasses, ice cliffs, and supraglacial meltwater channels are clearly visible in the map (Fig. 10c). Characteristic stripe-like patterns in the debris-covered area that are not recognisable in the visual orthophoto become apparent in the thermal orthophoto (Fig. 10d).

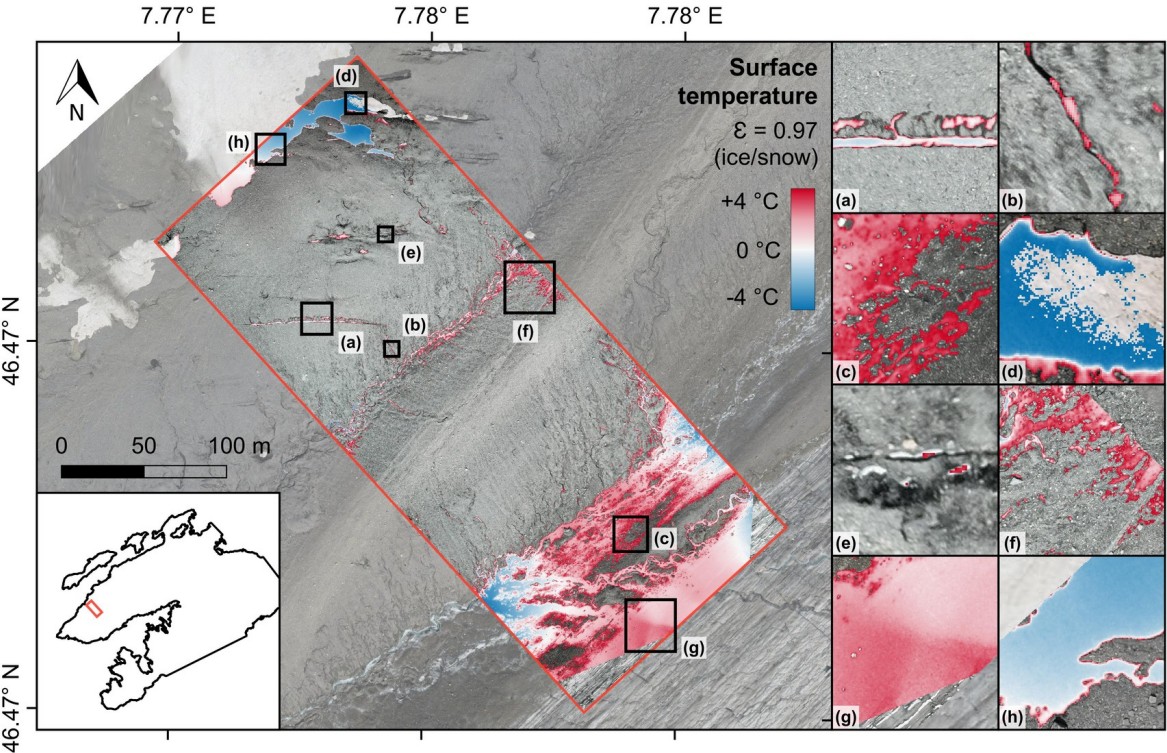

**Figure 11.** Corrected thermal orthophoto calculated with the emissivity of ice and snow ($\epsilon = 0.97$) to assess the accuracy and plausibility of the mapped surface temperatures. Only grid cells in the range from -4 °C to +4 °C are displayed. The enlarged map sections are discussed in the text. Background: UAV-based visual orthophoto from 14 September 2021. Note that besides the snowfield and ice cliffs only the glacier area outside the debris mask (see Fig. 12) is used for quantitative validation as the areas surrounding excerpts (c) and (f) comprise patchy debris (see Fig. 8).





Since snow and ice surfaces can be expected to be relatively close to the melting point on afternoons during the ablation season, such areas can serve as a reference for assessing the accuracy and plausibility of mapped glacier surface temperatures.

Throughout the entire surveyed glacier area, the deviation of the snow and ice surface temperatures from 0 °C is less than the measuring accuracy of the TIR camera (Fig. 11). A tiny part of the snow patch in the northernmost corner of the thermal orthophoto shows surface temperatures slightly below -4 °C (Fig. 11d) and some small ice cliffs and snow patches show surface temperatures slightly above 4 °C (Fig. 11e), but apart from that, the surface temperature in the debris-free area is in general in the range of -4 °C to 4 °C. The surface temperatures of debris-free ice range from 0.5 to 2.5 °C and of most larger ice cliffs

from -2 to 2 °C. The surface temperature of the larger ice cliffs in the centre of the survey area (Figure 11a,b) are close to the melting point (0.0 ± 0.5 °C). However, some artefacts originating from the photogrammetric processing become apparent (Fig. 11g). Moreover, a tendency towards higher deviations from the melting point can be observed towards the edges of the thermal orthophoto. Overall, the moderate deviation of the surface temperature from the melting point in the debris-free area (Pix4D: 0.4 ± 1.0 °C; ODM: 0.3 ± 1.0 °C) proves the suitability of the presented methodology for mapping glacier surface

temperature in high resolution and with adequate accuracy.

### 4.6 Spatial variations in supraglacial debris thickness

Since the supraglacial debris thickness maps were derived from the corrected thermal orthophoto, spatial variations in debris thickness and surface temperature correlate with each other (cf. Figs. 10 and 12). The debris thickness calculated with the empirical model shows a high spatial variability and ranges from a couple of millimeters up to 14 cm (Fig. 6). In general, the

debris layer becomes thicker towards the glacier margin below the southface of the Blüemlisalp (Fig. 12b). Beside this, the debris layer appears to be relatively thick in the elevated area between the parallel supraglacial meltwater streams (Fig. 12d). Close to the bare-ice surface, the debris cover tends to be very thin (<2 cm), apart from the supraglacial moraine at the debris-ice margin (labelled with "M" in Fig. 10). Individual spots of higher debris thickness indicate larger rocks or debris cones. A remarkable feature are the stripe-like patterns that are clearly visible in the debris thickness map (Fig. 12c,d), but poorly visible

in the visual orthophoto (Fig. 8). The underlying mechanism of formation is not known, but as the alternating stripes of thicker and thinner debris run parallel to the slightly inclined slopes, we assume that they are the result of frost sorting or rain- and meltwater drainage during the ablation season.

The results of the simple sensitivity experiment show that the (effective) thermal conductivity of the debris layer is a suitable parameter to calibrate the inverse surface energy balance model against in situ point measurements (Fig. 13). Since the thermal

conductivity is a very sensitive parameter in the model, it has a large influence on the spread of the modelled debris thickness. The greater the thermal conductivity, the thicker the debris (Fig. 14). Within the plausible range of potential thermal conductivities for a supraglacial debris layer (0.5-1.5 W m$^{-1}$ K$^{-1}$), the mean relative difference between the modelled and observed debris thickness varies between 1.2 and 2.6 cm. The model run with the smallest RMSE predicts a thermal conductivity in the order of 1.0 W m$^{-1}$ K$^{-1}$ for the debris layer on the Kanderfirn (Fig. 15). The remaining unexplained variance points to uncertainties

originating from the energy balance model itself, from the estimated model parameters (and their spatial variability) and from the mapped surface temperatures.



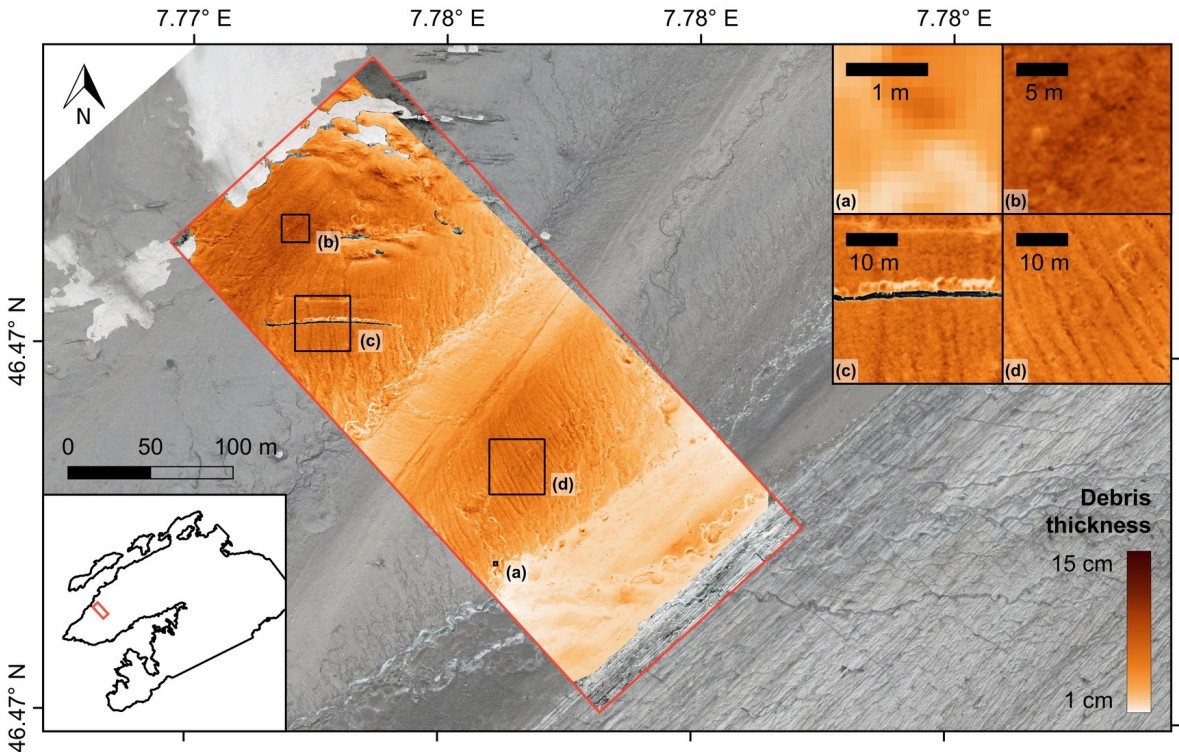

**Figure 12.** High-resolution map of supraglacial debris thickness calculated from the corrected thermal orthophoto using the empirical model (Eqn. 9). The map sections (a, b, c, d) are the same as in Fig. 8 and Fig. 10. Background: UAV-based visual orthophoto from 14 September 2021.

The debris thickness maps produced with the two different models (empirical and physical) agree well. They are generally very similar in terms of spatial patterns (Fig. 14). However, thickness deviations of up to ±2 cm are visible in some areas (Fig. C1). Compared to the empirical approach, the physical model seems to perform better in terms of predicting very thin and relatively thick debris. The maximum debris thickness predicted with the physical model is almost 25 cm (see Fig. 6) and therefore better reflects the height of individual boulders that are spread across the debris-covered area.

## 5 Discussion

### 5.1 Thermal imaging of debris-covered glaciers with UAVs

The aerial surveys on the Kanderfirn in September 2021 have shown that compact, low-cost UAVs equipped with a light-weight radiometric TIR camera facilitate the thermal imaging of debris-covered mountain glaciers in high-resolution and with adequate accuracy. Some of the limitations faced by previous studies that used oblique, field-based thermal imagery for high-resolution mapping of debris surface temperatures (Herreid, 2021; Tarca and Guglielmin, 2022), such as the restricted field of



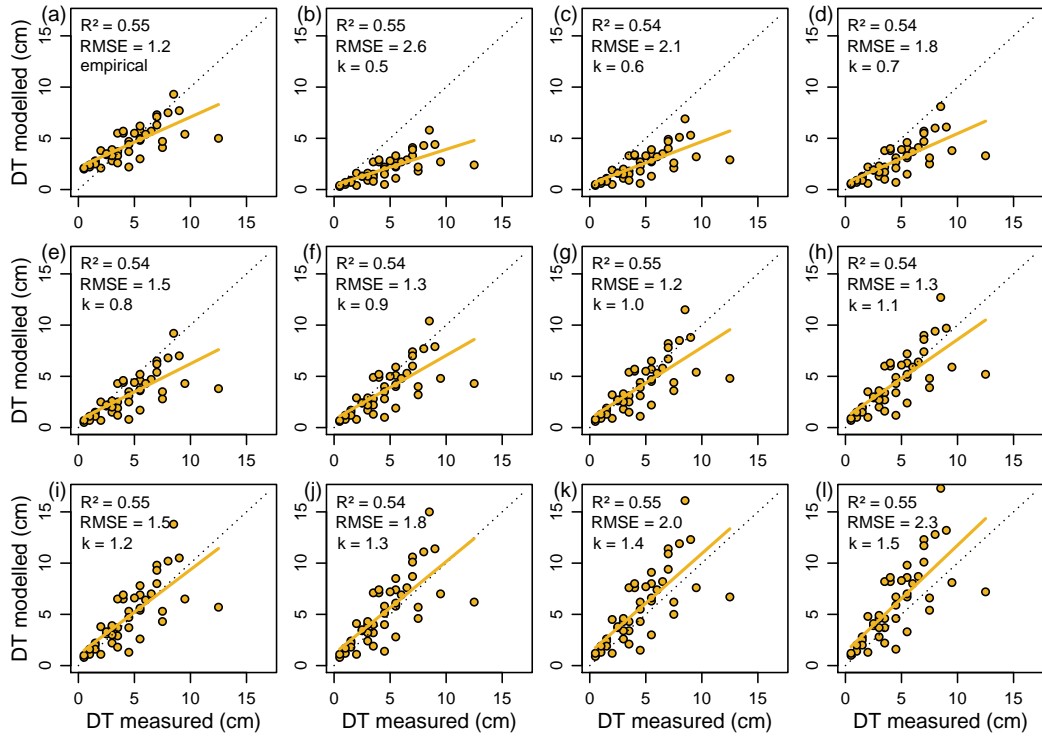

**Figure 13.** Measured vs. modelled debris thickness for different thermal conductivities ($0.1k \mid k \in [5..15]$) using the inverted glacier surface energy balance model (Eqn. 17) with the parameter values provided in Table 5.

view or variable ground sampling distance and atmospheric influence within one frame, can be overcome with the presented UAV-based mapping approach. However, other challenges arise from the deployment of UAVs and both terrestrial and aerial

thermal images require a proper calibration and validation.

One major drawback of using a light-weight rotary-wing UAV with a standalone TIR camera for aerial thermography at high altitude is the limited flight time due to the additional payload. With our customised UAV we achieved flight times of ca. 10 minutes and flight distances of max. 1000 m. Similar flight times are reported by Gök et al. (2022), who used a similar UAV-setup for debris thickness mapping on Glacier de Tsijiore Nouve in the Swiss Alps. Bisset et al. (2022) deployed

a more powerful quadcopter, a DJI Phantom 4, on the debris-covered Llaca Glacier in the Peruvian Andes. Since the UAV was operated at ca. 4,500 m, the achieved flight times were also relatively short (ca. 12 minutes). An alternative to rotary-wing UAVs are fixed-wing UAVs that facilitate much longer flight times and, thus, enable thermal imaging of larger glacier areas (Kraaijenbrink et al., 2018). However, sufficient space is needed for the safe landing of fixed-wing UAVs. Hence, their operation over rough debris-covered glacier tongues is de facto infeasible if there are no flat and smooth surfaces on or next

to the glacier. Tail-sitter UAVs that combine vertical take-off and landing with efficient forward flight could be a solution, but
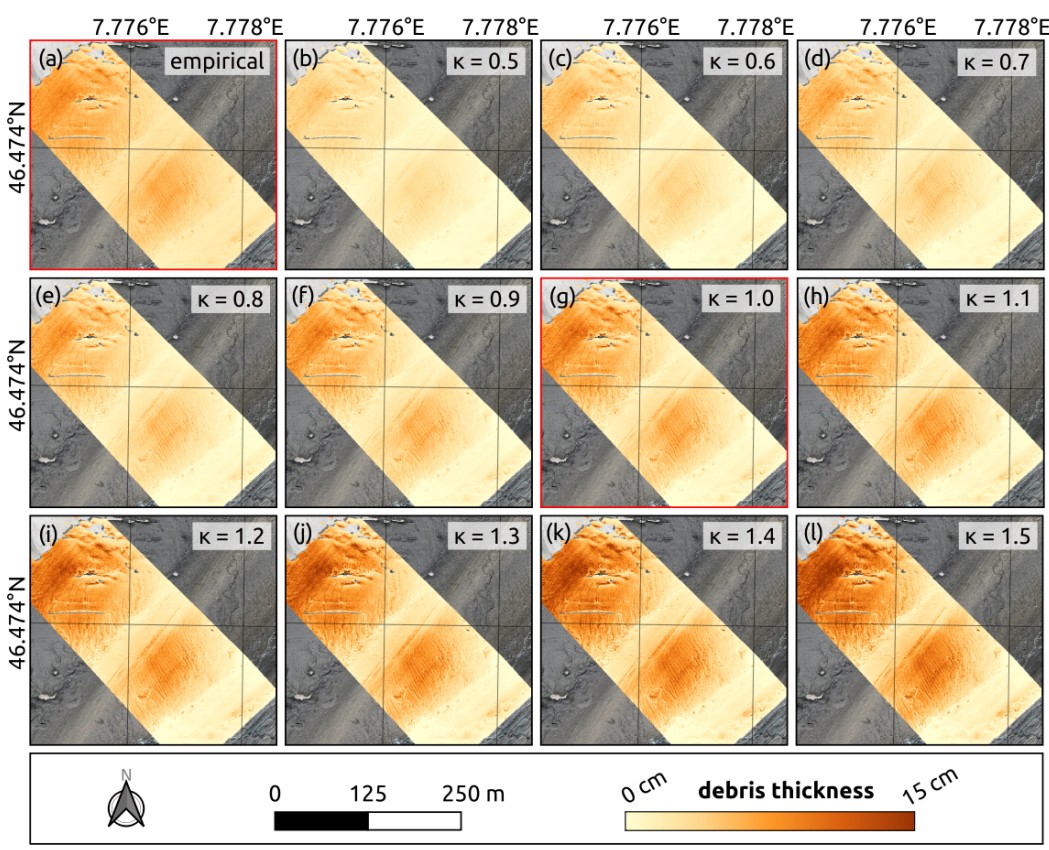

**Figure 14.** Impact of thermal conductivity on the spatial distribution of supraglacial debris thickness. The debris thickness maps were calculated from the corrected thermal orthophoto for different thermal conductivities $(0.1k \mid k \in [5..15])$ using the inverted glacier surface energy balance model (Eqn. 17) with the parameter values provided in Table 5. Note that the distributed debris thickness modelling with a thermal conductivity $(k)$ of 1.0 W m$^{-1}$ K$^{-1}$ (red frame) shows the highest agreement with the in situ measurements and the empirical map (red frame).

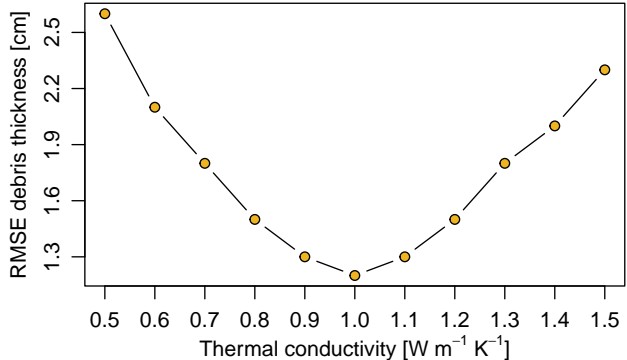

**Figure 15.** RMSE of modelled debris thickness (see Fig. 13) for different thermal conductivities ($0.1k \mid k \in [5..15]$) using the inverted glacier surface energy balance model (Eqn. 17) with the parameter values provided in Table 5.

their application in glaciology has so far been limited to calving glaciers in Greenland (e.g. Jouvet et al., 2018; van Dongen et al., 2021).

With regard to supraglacial debris thickness mapping, the timing of the UAV surveys and acquisition of thermal imagery matters. The two subsequent thermal UAV surveys on the Kanderfirn were performed in the early afternoon (between 13:51 and
14:40 CEST) as the surface of the debris layer should be closer to thermal equilibrium around midday than during the morning or evening hours when it heats up or cools down. Moreover, a high spatial heterogeneity in surface temperatures that reflects supraglacial debris thickness variations can be expected around midday (Bisset et al., 2022). The surface temperature range in the studied debris-covered area was indeed high (-4 to 30 °C, see Figs. 7 and 10) and facilitated the accurate determination of debris thickness. Based on the analysis of thermal orthophotos from different times of the day, Bisset et al. (2022) found
that the maximum spatial surface temperature variance of the debris layer on Llaca Glacier in the Peruvian Alps is reached at midday. The authors therefore also concluded that midday to early afternoon is a favourable time for UAV-based thermal imaging and debris thickness mapping. However, Herreid (2021) pointed out that the surface of a debris layer thicker than one decimetre can become nearly isothermic during the hottest hours of the day. The author therefore suggests for future studies to explore additionally the potential of thermal imaging during the early night, when thermal radiation emitted from a debris
layer may be well related to its thickness.

Since accurate surface temperatures are the prerequisite for reliable debris thickness mapping, radiometric thermal images collected with UAVs must be corrected for the effects of inherent sensors biases, thermal drift, atmospheric impacts, reflected thermal radiation and spatially varying emissivity (e.g. Ribeiro-Gomes et al., 2017; Kraaijenbrink et al., 2018; Virtue et al., 2021; Bisset et al., 2022; Gök et al., 2022). Low-cost uncooled microbolometers that are commonly used in UAV thermogra-
phy are notoriously sensitive to changing atmospheric and environmental conditions. Temperature fluctuations of the camera housing, lens or sensor, caused for example by changes in wind speed or air temperature, lead to a disequilibrium between the sensor and ambient air, which may considerably alter the surface temperature measurement. TIR cameras, such as the





FLIR VUE Pro R 640 used in this study and by Bisset et al. (2022), perform a flat field correction using the closed shutter at power-up and periodically during operation (the exact procedure is not disclosed by the manufacturer) to compensate partially

for the thermal drift during operation. However, the internal sensor calibration does not guarantee accurate surface temperature measurements. Gök et al. (2022) experienced considerable problems with the internal flat field correction of the FLIR Tau2 and Kraaijenbrink et al. (2018) observed a mean sensor bias in the order of 7 °C for the senseFly thermoMAP.

Some practical measures should be considered to minimise the temperature variations of the TIR camera and reduce thermal drift, although they cannot replace additional calibration and correction procedures. Thin altostratus or cirrostratus clouds, as

during the UAV surveys on the Kanderfirn, appear to be more favourable for mapping surface temperatures on debris-covered glaciers than clear skies as the debris layer heats up and cools down less quickly. Hence, it is less likely that the debris layer will become isothermal during day or night (Herreid, 2021). To avoid sudden heating and cooling of the sensor, it might be helpful to fly at low speed and place the TIR camera inside a wind and radiation shield instead of just mounting it at the underside of a rotary-wing UAV (Gök et al., 2022). Moreover, the camera needs time to stabilise after activation and adjust to

the ambient conditions (Pour et al., 2019; Kelly et al., 2019). Bisset et al. (2022) therefore turned on the camera early before takeoff and performed a couple of stabilising flight tracks before the actual survey. We switched on the camera a few minutes before take-off, but neglected additional flight tracks because of energy shortage.

In addition to practical measures, both calibration and correction procedures are required to obtain accurate surface temperatures of ice, snow and supraglacial debris. To account for inherent sensors biases, such as constant differences between actual

and recorded surface temperatures, Bisset et al. (2022) captured images of a blackbody at different temperatures with the TIR camera in the lab to determine a function that can be used to calibrate the acquired thermal images. This step was neglected in other UAV-based debris surface temperature mapping attempts, including ours (Kraaijenbrink et al., 2018; Gök et al., 2022). Stable black body temperatures captured in the lab could be used to create a filter that corrects for radial non-uniformities (known as 'vignetting') in temperature recordings caused by lens optics (Virtue et al., 2021). However, it appears that signal

averaging during photogrammetric processing mitigates the vignetting effects. While previous UAV-based studies used ice cliffs and debris-free glacier surfaces with an assumed skin temperature of 0 °C to calibrate thermal images and compensate indirectly for thermal drift and other effects (Kraaijenbrink et al., 2018; Bisset et al., 2022; Gök et al., 2022), our open-source pipeline allows for the correction of atmospheric interference and reflected thermal radiation. The problem of using ice cliffs on heavily debris-covered glaciers for the calibration and correction of thermal images is their small area and inclined surface,

which makes the recorded surface temperatures prone to radiative influences from the surrounding debris. A more reliable procedure to account for the thermal drift and atmospheric impacts would be the concurrent sounding of the atmospheric boundary layer above the glacier surface (see e.g. Hansche et al., 2023) during thermal UAV surveys and the use of a portable light-weight calibrator that can be attached to a TIR camera to enable consistent thermal measurements and increase the accuracy of absolute temperature recordings (e.g. Virtue et al., 2021). Additionally, calibration targets with a known emissivity

and temperature, distributed across the debris-covered area and measured with a differential GNSS device and high-precision hand-held TIR camera, would be useful (Kraaijenbrink et al., 2018; Bisset et al., 2022).



While debris-free glacier surfaces are suitable to assess the overall accuracy of mapped ice and snow temperatures, independent measurements are necessary to validate the mapped debris surface temperatures. In the absence of independent debris temperature measurements, as in the study of Gök et al. (2022), the reliability of the UAV-based debris surface temperature recordings remains unclear. Independent and accurate ground-based measurements of the surface temperature of varying supraglacial debris thicknesses can be performed with a hand-held TIR radiometer (Kraaijenbrink et al., 2018; Bisset et al., 2022). However, ground-based radiometric measurements are also subject to uncertainties (Aubry-Wake et al., 2015, 2018; Ribeiro-Gomes et al., 2017; Herreid, 2021). Furthermore, it is virtually impossible to manually measure surface temperatures at multiple locations on a debris-covered glacier within the time frame of a single UAV survey if only one expensive hand-held TIR radiometer is available. Tiny low-cost thermistor loggers as the ones used in this study (Section 3.3.2) are suitable to determine the near-surface debris temperature at various locations (see Figs. 1 and 9) and complement radiometric measurements. The thermodynamic and skin temperature of a debris layer are not the same, but they should nevertheless be highly correlated (Kraaijenbrink et al., 2018).

The validation of the mapped glacier surface temperatures on the Kanderfirn reveals that the presented open-source approach facilitates the generation of accurate thermal orthomosaics from raw thermal imagery acquired with a TIR camera mounted on a customised UAV. Measured near-surface debris temperatures correlated well with calibrated and corrected debris surface temperatures at the same point ($R^2$ = 0.71, RMSD = 3.5 °C). Larger deviations of more than 5 °C at three points originated probably from slight positional inaccuracies in both datasets. In the debris-free area, the mean deviation from the melting point (∼0 °C) was 0.4 ± 1.0 °C for the Pix4D and 0.3 ± 1.0 °C for the ODM thermal orthophoto. Similar to the first UAV-based high-resolution surface temperature maps of debris-covered glacier areas in the Himalaya, Andes and Alps (Kraaijenbrink et al., 2018; Bisset et al., 2022; Gök et al., 2022), our observations provide new insights into characteristic small-scale temperature variations of supraglacial debris. As the pronounced spatial surface temperature variations are associated with debris thickness variations, the created thermal orthophoto is a promising basis for high-resolution debris-thickness mapping.

## 5.2 Mapping supraglacial debris thickness in high resolution

As our results show, accurate and high-resolution supraglacial debris thickness maps can be generated from corrected UAV-based thermal orthophotos following the principles of previous studies dealing with satellite-based supraglacial debris thickness mapping (e.g. Mihalcea et al., 2008a, b; Foster et al., 2012; Juen et al., 2014; Rounce and McKinney, 2014; Groos et al., 2017; Rounce et al., 2018, 2021). While both the empirical model and the inverse surface energy balance model provide reasonable estimates of the debris thickness distribution, the Kanderfirn results suggest that the physical model better captures the full range of debris thickness (Fig. 6). Gök et al. (2022) found that their empirical model predicts slightly thicker debris than their physical model, but this observation might be biased by the use of a literature value for the site-specific thermal conductivity in the physical model. In the absence of additional site-specific data regarding debris properties and meteorological conditions, the empirical approach is appropriate. A major advantage of using an inverse surface energy balance model, however, is that it can theoretically account for spatial variations in the meteorological conditions and debris properties (such as albedo, thermal conductivity, and surface topography), which influence the debris thickness prediction (Bisset et al., 2022; Gök et al., 2022).



A prerequisite for simulating debris thickness from thermal UAV imagery with an inverse energy balance model are accurate information on the debris properties and the meteorological conditions during the UAV surveys. Similar to Bisset et al. (2022), we used meteorological data from weather stations in the surrounding area to estimate the mean air temperature, relative humidity, and incoming short- and longwave radiation at the study site during the thermal imaging. Since our UAV surveys

were completed in less than one hour, we did not account for any temporal variations. Uncorrected ERA5 climate reanalysis data with a horizontal resolution of 0.1° x 0.1° as used by Gök et al. (2022) seem unfavourable as they fail to characterise the heterogeneous meteorological conditions in mountainous terrain (e.g. Khadka et al., 2022). In an ideal setup, a weather station on the glacier in conjunction with atmospheric profiling (e.g. Hansche et al., 2023) during the thermal UAV surveys would provide the necessary data to model the spatial and temporal distribution of the individual meteorological variables

accurately across the debris-covered area (Bisset et al., 2022; Gök et al., 2022). Two highly sensitive debris parameters in surface energy balance models are typically albedo and effective thermal conductivity (e.g. Nicholson and Benn, 2006; Rounce and McKinney, 2014; Groos et al., 2017; Bisset et al., 2022; Gök et al., 2022). Bisset et al. (2022) modelled the thermal conductivity at one site on Llaca Glacier using debris temperature measured at different depths. As the installation of multiple thermistors at varying depths is impractical in thin debris, we calibrated this parameter against in situ thickness observations.

While distributed modelling of debris thickness remains an unresolved issue, debris albedo could be mapped in high resolution using visual UAV imagery (e.g. Ryan et al., 2017).

The validation of the created debris thickness map at 22 points demonstrates that the empirical model and calibrated physical model are able to reproduce independent field observations. The RMSE of both models is 1.2 cm. Comparing the accuracy achieved for the Kanderfirn with that of recently published high-resolution debris thickness maps of Llaca Glacier in the

Peruvian Andes and Glacier de Tsijiore Nouve in the Swiss Alps is of limited value as not only the mean debris thickness of the three glaciers differs, but also the number and reliability of field observations. On Llaca Glacier, only three debris thickness measurements ranging from 3 to 26 cm were available. At all three points, the deviation between the modelled and measured thickness was less than 2 cm (see Fig. 5 in Bisset et al., 2022). On Glacier de Tsijiore Nouve, debris thickness was measured at 90 locations, but the position was only determined with a handheld GPS device (Gök et al., 2022). It is therefore unclear

whether the high RMSE of 6 to 8 cm reported for the simulated debris thickness originates mainly from uncertainties in the thermal imaging, model parametrisation or positional measurements.

The high-resolution mapping on the Kanderfirn reveals typical small-scale features and remarkable debris thickness patterns, such as debris cones and parallel stripes (Fig. 12), that cannot be resolved spatially by the TIR sensors of current-generation satellites. A simple comparison of the high-resolution map and additional field observations from the Kanderfirn with the

global debris thickness dataset of Rounce et al. (2021) shows that the global debris mask comprises less than a quarter of the entire debris-covered area of the Kanderfirn. Moreover, the modelled debris thicknesses derived from satellite data seem to overestimate considerably the true thickness. A similar observation has been made by Bisset et al. (2022). The use of the global debris thickness dataset in glacier surface mass or energy balance models might result in a serious underestimation of ablation, at least on individual glaciers. However, a proper accuracy assessment of satellite-based debris thickness estimates and sub-

debris ice melt models would require more systematic measuring and mapping efforts of debris thickness and sub-debris ice



melt rates on various debris-covered glaciers in different regions. This could be achieved by combining classical in situ debris thickness and ablation measurement techniques with novel UAV-based mapping approaches. Besides thermal imaging, other promising approaches as the derivation of debris thicknesses from UAV-based elevation change data using an inverted Østrem curve should be further developed and tested (Westoby et al., 2020; Steiner et al., 2021).

## 6 Conclusions

We have presented a low-cost and open-source approach for high-resolution mapping of supraglacial debris thickness using UAV-based infrared thermography. The modified low-cost UAV equipped with a light-weight radiometric TIR camera facilitates the thermal imaging of smaller debris-covered areas and paves the way for glacier-wide high-resolution debris thickness mapping. Since the flight time of rotary-wing UAVs is a limiting factor at high altitudes, tail-sitter UAVs that combine vertical
take-off and landing with efficient forward flight are envisaged as a technical solution for future applications on debris-covered glaciers. Accurate raw thermal orthomosaics can be generated from the captured radiometric UAV imagery using the open-source pipeline developed around the OpenDroneMap photogrammetry software. The great advantage of the workflow is that user-specific calibration and correction procedures can be easily applied to a raw thermal orthophoto to obtain accurate surface temperature maps. In the debris-free area of the Kanderfirn, an accuracy of $0.4 \pm 1.0$ °C was achieved. However, more so-
phisticated calibration and correction procedures are needed for longer thermal UAV surveys and glacier-wide debris thickness mapping. We suggest that future studies perform parallel atmospheric sounding during the UAV surveys and take advantage of a portable light-weight calibrator that can be attached to a TIR camera to enable consistent thermal measurements and increase the accuracy of absolute temperature recordings. Our results from the Kanderfirn show that accurate debris thickness maps (RMSE of 1.2 cm) at decimetre-resolution can be derived from a corrected thermal orthophotos using an empirical function or
an inverted sub-debris ice melt model. These maps provide more accurate and realistic estimates of the debris thickness distribution on individual mountain glaciers than products derived from satellite data. Combining UAV-based infrared thermography and debris thickness mapping with in situ measurements and numerical modelling therefore opens up new opportunities for monitoring the current state of debris-covered glaciers and improving the projection of their future evolution.

*Code and data availability.* The complete dataset (including visual and thermal UAV images, in situ measurements, visual and raw thermal
orthophotos, and surface temperature and debris thickness maps), a QGIS project, and the open-source pipeline for data processing and debris thickness modelling can be downloaded via the open-access repository Zenodo: https:doi.org/10.5281/zenodo.7692542 (Messmer and Groos, 2023).





# Appendix A: In situ measurements

**Table A1.** Debris thickness ($h_d$) measured at 43 points in the study area (see Fig. 1 and Section 3.3.1).

| ID | $h_d$ [cm] | Lon [°E] | Lat [°N] | Ele [m a.s.l.] | ID | $h_d$ [cm] | Lon [°E] | Lat [°N] | Ele [m a.s.l.] |
|---|---|---|---|---|---|---|---|---|---|
| d1 | 8.0 | 7.7744059 | 46.4746226 | 2464.2 | d31 | 5.5 | 7.7771062 | 46.4741961 | 2463.8 |
| d2 | 12.5 | 7.7752696 | 46.4750962 | 2473.5 | d32 | 3.0 | 7.7762456 | 46.4731511 | 2448.3 |
| d3 | 5.5 | 7.7750863 | 46.4743069 | 2462.4 | d33 | 3.5 | 7.7766598 | 46.4734852 | 2453.0 |
| d4 | 7.5 | 7.7758346 | 46.4745445 | 2464.1 | d34 | 3.0 | 7.7772580 | 46.4736869 | 2450.8 |
| d5 | 4.5 | 7.7752237 | 46.4737580 | 2449.0 | d35 | 4.0 | 7.7767208 | 46.4732632 | 2445.8 |
| d6 | 3.5 | 7.7767673 | 46.4741753 | 2460.4 | d36 | 2.0 | 7.7769406 | 46.4730988 | 2439.1 |
| d7 | 5.5 | 7.7762866 | 46.4735537 | 2456.7 | d37 | 1.5 | 7.7775412 | 46.4733837 | 2444.1 |
| d8 | 5.0 | 7.7771217 | 46.4737229 | 2453.1 | d38 | 4.5 | 7.7777958 | 46.4737172 | 2448.6 |
| d9 | 2.5 | 7.7768808 | 46.4730179 | 2437.7 | d39 | 0.5 | 7.7775861 | 46.4730375 | 2440.8 |
| d10 | 1.0 | 7.7778770 | 46.4731402 | 2444.0 | d40 | 3.0 | 7.7780300 | 46.4730089 | 2445.9 |
| d11 | 0.5 | 7.7774956 | 46.4724581 | 2442.3 | d41 | 1.5 | 7.7777237 | 46.4727502 | 2442.6 |
| d12 | 1.0 | 7.7783992 | 46.4731051 | 2446.0 | d42 | 4.5 | 7.7760154 | 46.4743723 | 2460.6 |
| d13 | 9.0 | 7.7746947 | 46.4747025 | 2466.9 | d43 | 6.0 | 7.7767335 | 46.4737731 | 2458.7 |
| d14 | 9.5 | 7.7748188 | 46.4749674 | 2469.5 | | | | | |
| d15 | 7.0 | 7.7753494 | 46.4748599 | 2470.3 | | | | | |
| d16 | 7.0 | 7.7751051 | 46.4746101 | 2465.4 | | | | | |
| d17 | 5.5 | 7.7748818 | 46.4741528 | 2460.3 | | | | | |
| d18 | 8.5 | 7.7746787 | 46.4744966 | 2464.4 | | | | | |
| d19 | 7.5 | 7.7757735 | 46.4748341 | 2469.8 | | | | | |
| d20 | 3.5 | 7.7756599 | 46.4745175 | 2464.4 | | | | | |
| d21 | 6.5 | 7.7753992 | 46.4743252 | 2461.6 | | | | | |
| d22 | 7.0 | 7.7752594 | 46.4739935 | 2453.8 | | | | | |
| d23 | 4.0 | 7.7756924 | 46.4742053 | 2457.0 | | | | | |
| d24 | 2.5 | 7.7755870 | 46.4738448 | 2448.7 | | | | | |
| d25 | 5.5 | 7.7758143 | 46.4741450 | 2455.3 | | | | | |
| d26 | 1.5 | 7.7764351 | 46.4744113 | 2460.0 | | | | | |
| d27 | 2.0 | 7.7760505 | 46.4739979 | 2452.1 | | | | | |
| d28 | 3.5 | 7.7760377 | 46.4737716 | 2454.1 | | | | | |
| d29 | 4.5 | 7.7760882 | 46.4735067 | 2455.6 | | | | | |
| d30 | 3.0 | 7.7765535 | 46.4740443 | 2458.8 | | | | | |





**Table A2.** Mean debris temperature (at a depth of ca. 0.5-1.0 cm below the surface of the debris layer) during the two UAV surveys, measured with the tiny temperature loggers (see Fig. 1 and Section 3.3.2).

| Logger | $T_{debris}$ [°C] | $h_d$ [cm] | Longitude [°E] | Latitude [°N] | Elevation [m a.s.l.] |
|---|---|---|---|---|---|
| t1 | 22.3 | 8.0 | 7.7744059 | 46.4746226 | 2464.2 |
| t2 | 21.9 | 12.5 | 7.7752696 | 46.4750962 | 2473.5 |
| t3 | 21.6 | 5.5 | 7.7750863 | 46.4743065 | 2464.1 |
| t4 | 16.1 | 7.5 | 7.7758346 | 46.4745445 | 2462.4 |
| t5 | 19.5 | 4.5 | 7.7752237 | 46.4737580 | 2449.0 |
| t6 | 14.3 | 3.5 | 7.7767672 | 46.4741753 | 2460.4 |
| t7 | 21.8 | 5.5 | 7.7762866 | 46.4735537 | 2456.7 |
| t8 | 17.8 | 5.0 | 7.7771217 | 46.4737229 | 2453.1 |
| t9 | 10.4 | 2.5 | 7.7768808 | 46.4730179 | 2437.7 |
| t10 | 11.9 | 1.0 | 7.7778770 | 46.4731402 | 2444.0 |
| t11 | 5.0 | 0.5 | 7.7774956 | 46.4724581 | 2446.0 |
| t12 | 4.5 | 1.0 | 7.7783992 | 46.4731051 | 2442.3 |

**Table A3.** Positional accuracy of the visual orthophoto at each GCP measured with the differential GNSS device (see Fig. 1 and Section 3.2).

| GCP | XY error [cm] | Measurement uncertainty [cm] | Longitude [°E] | Latitude [°N] | Elevation [m a.s.l.] |
|---|---|---|---|---|---|
| vGCP1 | 10 | 120 | 7.7744276 | 46.4746185 | 2464.1 |
| vGCP2 | 5 | 30 | 7.7752557 | 46.4750917 | 2473.2 |
| vGCP3 | 10 | 10 | 7.7750979 | 46.4743114 | 2462.4 |
| vGCP4 | 3 | 10 | 7.7758416 | 46.4745526 | 2464.1 |
| vGCP5 | 4 | 10 | 7.7752381 | 46.4737511 | 2448.8 |
| vGCP6 | 17 | 10 | 7.7767582 | 46.4741764 | 2460.3 |
| vGCP7 | 6 | 10 | 7.7763003 | 46.4735587 | 2456.7 |
| vGCP8 | 6 | 10 | 7.7771372 | 46.4737282 | 2453.0 |
| vGCP9 | 9 | 10 | 7.7768879 | 46.4730228 | 2437.8 |
| vGCP10 | 7 | 2 | 7.7778692 | 46.4731353 | 2443.9 |
| vGCP11 | 18 | 20 | 7.7776062 | 46.4723808 | 2443.5 |
| vGCP12 | 3 | 10 | 7.7786165 | 46.4729375 | 2448.6 |
| RMSE | 8 | | | | |





**Table A4.** Positional accuracy of the thermal orthophoto at each GCP measured with the differential GNSS device (see Fig. 1 and Section 3.2).

| GCP | XY error [cm] | Measurement uncertainty [cm] | Longitude [°E] | Latitude [°N] | Elevation [m a.s.l.] |
|---|---|---|---|---|---|
| tGCP1 | 12 | 110 | 7.7744005 | 46.4746085 | 2464.0 |
| tGCP2 | 22 | 50 | 7.7752344 | 46.4750881 | 2472.9 |
| tGCP3 | 7 | 10 | 7.7751199 | 46.4743199 | 2462.5 |
| tGCP4 | 7 | 10 | 7.7758623 | 46.4745579 | 2464.2 |
| tGCP5 | 11 | 10 | 7.7752625 | 46.4737512 | 2448.7 |
| tGCP6 | 7 | 10 | 7.7767335 | 46.4741733 | 2460.1 |
| tGCP7 | 5 | 10 | 7.7763206 | 46.4735528 | 2456.5 |
| tGCP8 | 10 | 2 | 7.7771517 | 46.4737350 | 2452.9 |
| tGCP9 | 6 | 2 | 7.7769003 | 46.4730335 | 2437.9 |
| tGCP10 | 12 | 10 | 7.7778508 | 46.4731243 | 2443.8 |
| tGCP11 | 11 | 10 | 7.7775906 | 46.4723888 | 2443.3 |
| RMSE | 9 | | | | |



## Appendix B:  Thermal orthophoto comparison

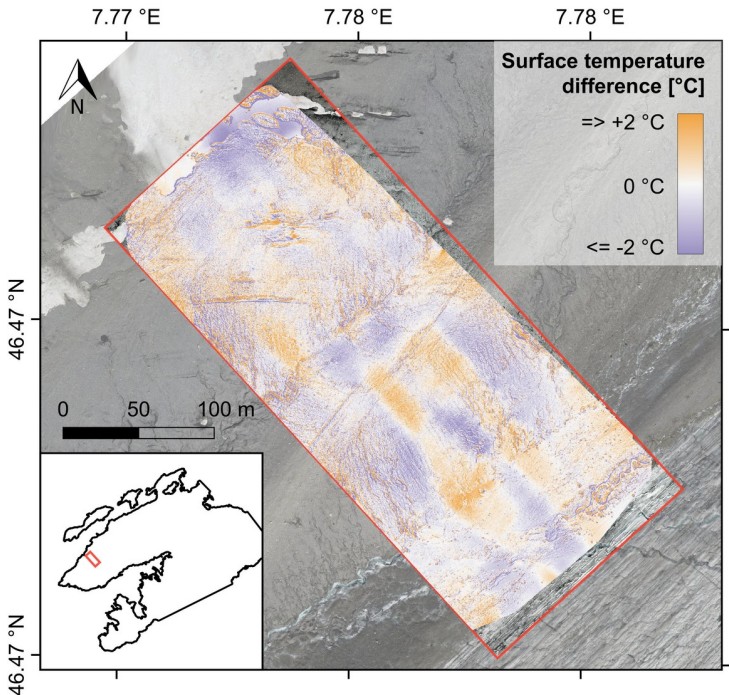

**Figure B1.** Difference between the surface temperature map derived from the raw thermal orthophoto generated with the open-source software ODM and the one derived from the raw thermal orthophoto generated with the proprietary software Pix4Dmapper. Positive deviations indicate relatively higher surface temperatures in the Pix4Dmapper thermal orthophoto and vice versa.



**Appendix C:  Debris thickness map comparison**

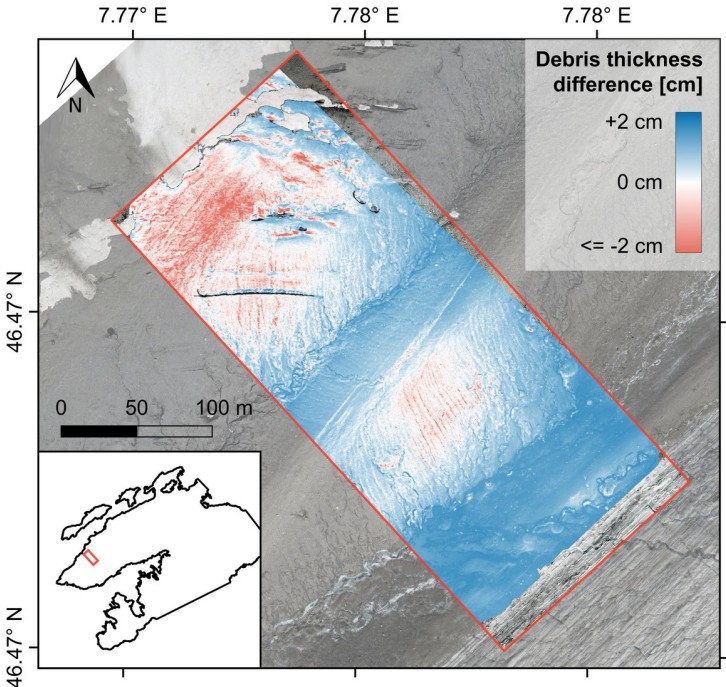

**Figure C1.** Difference between the debris thickness map calculated with the empirical model (Eqn. 9) and the one calculated with the inverse surface energy balance model ($k$ = 1.0 W m$^{-1}$ K$^{-1}$). Positive deviations (blue) indicate relatively higher debris thickness estimates in the empirical map and vice versa.





*Author contributions.* ARG designed the study, developed the customised mounting of the TIR camera on the quadcopter and implemented the surface energy-balance model. JM carried out the fieldwork, performed the UAV surveys and processed most of the data. Both authors contributed equally to the development of the open-source pipeline, writing of the manuscript and generation of the figures.

*Competing interests.* Both authors declare that they have no competing interests.

*Disclaimer.* Publisher's note: Copernicus Publications remains neutral with regard to jurisdictional claims in published maps and institutional affiliations.

*Acknowledgements.* The expenses for the UAV surveys and field work on the Kanderfirn were covered by the Institute of Geography of the University of Bern. We thank Peter Leiser for helping with the customised mounting of the TIR camera on the quadcopter and Christoph Mayer for contributing to the implementation of the theoretical sub-debris ice melt model. We also very much appreciate the support of
Rebecca Siebeneicher, Shayna Lindemann and Nicolas Brand during field work. The Swiss Federal Office of Meteorology and Climatology (MeteoSwiss) and the WSL Institute for Snow and Avalanche Research (SLF) kindly provided the meteorological data from Gandegg, Grächen, Fisistock, and Jungfraujoch.



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
