# Peer review of "A low-cost and open-source approach for supraglacial debris thickness mapping using UAV-based infrared thermography"

_The Cryosphere, 2023_

## Author Comment (AC1)

**General response**

We thank the editor Kang Yang for obtaining two valuable reviews and Sam Herreid and the anonymous reviewer for their thorough and constructive comments on our manuscript. On the following pages, we address the reviewers' comments point by point. The reviewers' comments are highlighted in grey. We hope that our responses will qualify us to submit a revised version of the manuscript.

**Response to Referee Comment 1 (RC1)**

The article "A low-cost and open-source approach for supraglacial debris thickness mapping using UAV-based infrared thermography" by Messmer and Groos provides an approach to bypass proprietary software to derive accurate surface temperature measurements and solve for debris thickness. The field experiment was thoughtfully set up and the article was a pleasure to read. It goes slightly against the title that the bulk of the analysis was conducted on data processed with proprietary software. This paper is an addition to a growing set of similar papers (mostly well cited within) which support the underlying premise of a coupled signal between surface temperature and debris thickness. This study, however, conveniently avoids the trickier aspects of this line of research by limiting the study site to a location where debris cover is generally thin, <15 cm, where the signal is expected to be strong. The spatial domain of these similar papers is also seemingly stuck to a small swath of a small glacier. Here, the authors put forth some good suggestions on how to break out of this limited domain in a UAV context, but they do not explore them. To further focus the paper, I think the discussion section could be shortened / organized with more subheaders. I also think it would improve the impact of the paper to add a specific discussion section on scaling the method and method repeatability. The in-line comments below are mostly minor.

Thanks for the detailed and constructive review. Following the suggestions above, we have made the following basic changes to the manuscript:

1) For the sake of consistency, we performed all processing steps with open-source software and used the results from the proprietary software only for comparison.

2) We restructured and organised the discussion using subheaders.

3) We added a discussion section on the scaling and repeatability of the method.

4) We performed a simple sensitivity analysis of the inverse surface energy balance model (as suggested by the other reviewer, see NEW Fig. 13).

Considering the complexity of accurate UAV-based thermal imaging in the mountains and high-resolution supraglacial debris-thickness mapping, we think it is reasonable to choose a simple setup for a proof-of-concept study. We have managed to extend the survey area considerably in a follow-up experiment and elaborate now in more detail on the scaling of the method in the revised discussion.

We do not agree that detecting thicker debris (i.e. >50 cm) should be the benchmark for a debris

thickness mapping technique. First, relative thin debris (i.e. <15 cm) predominates on most glaciers and accounts usually for more than 50 % of the total debris-covered area (see Rounce et al., 2021; McCarthy et al., 2022). Second, the melt rate asymptotically approaches a low rate of melt when the debris becomes thicker than ~50 cm (Østrem, 1959; Evatt et al., 2015). While we generally agree that the suitability of the presented methodology needs to be demonstrated for debris thicker than 15 cm in a follow-up study, we think that at least in the context of modelling the smb and evolution of debris-covered glaciers, larger mapping uncertainties in the thicker debris-covered areas are tolerable.

Abstract

L4: Perhaps less accurate and lower resolution, but it seems a little remiss to not mention Rounce et al., 2021 re: a global map of debris cover.

We refer to the global debris thickness map of Rounce et al. (2021) at various sections in the text. Here we merely emphasise that there is a global lack of in-situ debris thickness measurements and observations on small-scale variations in debris thickness and sub-debris ice melt rates, leading to considerable uncertainties in the modelling of debris-covered glaciers. The summarising table S1 of Rounce et al. (2021) supports our statement that (published) in-situ measurements of debris thickness and sub-debris ice melt rates are rare (limited to less than 50 glaciers worldwide).

L5: Maybe it's "describe" the customization of a low cost UAV and "present" a complete open source..

modified

L8: Is it still raw though? I assume during the SfM step there was some statistical resampling. I'm fine if you still call it raw, but to me that is strictly the values of the individual radiometric images.

This is a nuance, but we agree and replaced "raw thermal" by "radiometric" throughout the text

L9: provided the dt spanning the set of images is reasonably low. I'm sure you say this in the text.

This sentence only states that distinct surface temperature variations were found across the surveyed debris-covered area. The relationship between dt and Ts is not meant here. We revised the sentence: "The thermal orhthophoto reveals distinct spatial variations in surface temperature across the surveyed debris-covered area."

L13: using which method, empirical or inverse EB? Or both?

Revised: "...can be mapped with high accuracy using an empirical or physical model."

L15: is this still for a 0-15 cm class? Or for the whole DC area?

Debris thicknesses above 15 cm are very rare or absent on the Kanderfirn. We therefore assume that the RMSE of 1.3 cm (updated using the results from the open-source pipeline) calculated as the difference between modelled and observed debris thicknesses is representative for the entire debris-covered area.

L17: "paves the way" Does it? I'm not sure when we'll have regional, not to mention glacier-wide, UAV coverage and moving up to fixed wing or helicopter increases the cost by quite a bit.

Yes, we are convinced that the approach presented here and in other publications (Bisset et al., 2022; Gök et al., 2022) paves the way for comprehensive debris thickness mapping (at least of mountain glaciers, maybe not of large debris-covered outlet glaciers) with UAVs in the order of a couple of square kilometres (we explicitly do not refer to regional attempts), although of course several practical and methodological challenges (related inter alia to spatio-temporal variations in the meteorological conditions and debris properties) are involved. Current low-cost fixed-wing or hybrid UAVs are already capable of surveying extensive areas in glacierised environments (e.g. Jouvet et al., 2019: https://doi.org/10.3389/feart.2019.00206). We elaborate more on the scaling of this approach in the revised discussion.

Revised sentence: "The presented approach paves the way for comprehensive high-resolution supraglacial debris thickness mapping and opens up new opportunities for more accurate monitoring and modelling of debris-covered glaciers."

Introduction

L44: Rounce et al., 2023 seems to temper this urgency, at least in terms of global sea level predictions. "The inclusion of debris thus delays mass loss over the century especially at local scales but has little impact on sea level rise and the number of glaciers lost by 2100." Regionally, and for addressing problems on shorter timescales, the urgency seems to remain warranted.

We do not refer to sea level rise here. We appreciate the work of Rounce and colleagues, but one might question how reliable the global map and simulations are with respect to the properties and evolution of debris-covered glaciers in a certain region. Bisset et al. (2022) indicate that the global debris thickness map of Rounce et al. (2021) derived from satellite data might overestimate the debris thickness (at least for individual glaciers), with considerable impacts on the modelled sub-debris melt rates. As the meltwater from debris-covered glaciers is an important fresh water source in different regions, we think the addressed urgency regarding accurate information on debris thickness distribution is warranted.

L53: Consider saying more about the low accuracy since this is key motivation for finescale work such as this. You can more explicitly constrain "low accuracy" with repeatability arguments and the often stated ~0.5 m detection limit. From Rounce et al. 2021, we now know where we can anticipate this limit to be met.

We revised the section as follows: "However, satellite remote sensing has the drawback that the acquisition time, viewing angle and atmospheric conditions cannot be controlled by the end-user and might not be in favour of debris-thickness mapping (e.g. Herreid, 2021). Debris thickness maps based on satellite remote sensing data usually capture general debris thickness patterns in the ablation zone, but cannot resolve the small-scale debris thickness variability and the presence of melt hotspots such as supraglacial ice cliffs and ponds due to their relatively coarse spatial resolution. Since the relationship between debris thickness and sub-debris ice melt is non-linear (Østrem, 1959) and melt spots are not resolved, the mean debris thickness per pixel derived from relatively coarse satellite data seems not well suited to simulate sub-debris ice melt rates."

L55: Higher resolution mapping techniques are required to do what? Brute force solve the problem with e.g. aerial surveys, or inform satellite based methodology either with sensors in orbit today or wait for civilian satellite surface temperature data to become higher resolution?

As we outline before, the mean debris thickness per pixel derived from relatively coarse satellite data seems not well suited for simulating sub-debris ice melt rates as the small-scale debris-thickness variability and melt hot spots such as cliffs and ponds are not resolved. So if we aim to quantify/simulate the ablation and meltwater contribution of debris-covered glaciers in a more accurate and robust manner (for example in the context of local climate adaption strategies or water management), high-resolution debris-thickness mapping techniques are required. We added a section in the discussion to outline how in-situ, UAV and satellite data might be combined in the future for supraglacial debris thickness mapping on larger scales.

L59: See also:

Gök, Deniz Tobias, Dirk Scherler, and Leif Stefan Anderson. "High-resolution debris-cover mapping using UAV-derived thermal imagery: limits and opportunities." The Cryosphere 17.3 (2023): 1165-1184.

Aubry-Wake, Caroline, et al. "Using ground-based thermal imagery to estimate debris thickness over glacial ice: fieldwork considerations to improve the effectiveness." Journal of Glaciology 69.274 (2023): 353-369.

We quote Gök et al. (2022) a few sentences further below. Thanks for the hint, we were not aware of the latest publication of Aubry-Wake et al. (2023). The paper is now quoted.

L61: The nadir camera angle is without a doubt a benefit, and a UAV can capture a wider swath of a glacier. But still, the narrow swaths of high resolution thermal data presented here and in all other similar studies I have seen, the spatial coverage is not operative for studies beyond proof of concept. Further, there are radiometric resolution trade-offs between a heavy ground based camera and those that meet the payload of a UAV. I'm not making an argument for more ground based studies, I'm just saying proof of concept studies might benefit from the better (heavier) sensors and the spatial gains from today's UAVs are not orders of magnitude better.

We fully agree that the thermal orthophoto presented in this study is not orders of magnitude larger than the area that can be surveyed with ground-based infrared thermography. However, we think that a small area is reasonable for a proof of concept and the presentation of a complex processing pipeline. The challenges related to the thermal imaging of larger areas with (fixed-wing) UAVs is going to be addressed in a follow-up paper. However, the general feasibility of UAV-based thermal imaging of larger areas has already been proven in previous studies… (e.g. Kraaijenbrink et al., 2018).

L66 Or hinting at its futility? For a method to be robust, the cannon of similar papers will need to all start converging on good news and repeatability or the general concept should probably be sidelined.

Mapping supraglacial debris with UAVs is tricky (both from a practical and physical point of view), but this study and the studies of Bisset et al. (2022) and Gök et al. (2022) demonstrate that

it is  feasible in principle. We do not understand why a new technique should not be further developed or even sidelined simply because of technical issues reported in a single study.

L69: I assume it would work on thermal imagery acquired by any means?

That's true. Modified.

L73: debris [surface] temperature.

Debris temperature and not debris surface temperature is indeed meant here.

Study Area

L87: The debris-covered area of the Kanderfirn is already very small, on a bigger debris covered glacier, say, the terminus of Bering Glacier in Alaska as an extreme, how would you decide where to survey? Is there any path for scalability with this method?

This is an interesting question. While high-endurance UAVs might be able to survey large areas such as the Bering Glacier in Alaska and survey its surface temperature in the future, one might argue that airborne or helicopter-borne missions would be more suitable. Independent of the vehicle, accounting for the varying atmospheric conditions and debris properties across such a large area remains a big challenge. To assess the small-scale variability of debris thickness and quantify its impact on glacial ablation of such large glaciers, UAV-based thermal imaging at a limited number of sites along the flowline and perpendicular to the flowline might be an option. The created high-resolution thermal orthophotos and debris thickness maps might then serve as a reference/basis for the validation, calibration or downscaling of glacier-wide debris thickness maps based on satellite data.

Data and Methods

L117: Recommended in the literature or from this study?

Recommended in the literature. Sentence was modified accordingly: "...a lateral image overlap of at least 60 % is common in UAV photogrammetry (e.g. Kraaijenbrink et al. 2018; Jouvet et al., 2019)".

Table 1: The smaller TIR area surveyed from the same flight must mean that the overlap was less than for the visible imagery? This difference is due to the view angle? Is the 75% from L118 for RBG or TIR?

This is true. The lateral overlap of the TIR imagery was 75 %, while that of the RGB imagery was even larger. We modified the sentence accordingly: "...resulting in a lateral image overlap of 75~\% for the TIR imagery and even more for the RGB imagery."

L139: Literally random, as in picked by an algorithm before going into the field, or just random sampling while you were in the field?

The sentence was modified: "...that were randomly distributed across the survey area while being in the field."

L148: I'm not sure if it was the correct decision to put the temperature loggers under a rock to shield it from direct radiation since the thermal camera will be measuring the skin temperature and thus include that very warm signal. It's not unusual to measure 30 degree debris surface temperature on a cool but sunny summer day (as you know from your Fig. 10). It helps a little bit that it was slightly overcast (L113), Herreid, 2021 found overcast conditions to be preferable for TIR / debris thickness / sub-debris melt measurements. Studies like this one that look at a debris cover internal temperature profile along with contact thermistor and TIR surface temperature measurements show smooth and predictable diurnal signals just below the surface and lower (below your shale stone) and more chaotic signals for both TIR and contact thermistors at the very surface (chaotic as a function of clouds passing). Both have the scientific uses, I think in your case the skin measurement is correct, but what you collected should still be close.

We shifted the essential sentence from the Results section here: "Since the tiny temperature loggers recorded the debris temperature at shallow depths and the TIR camera mounted on the UAV measured the skin temperature of the debris layer, the in situ measurements and mapped surface temperatures cannot be expected to match exactly. However, a cross-comparison makes it possible to detect a potential warm or cold bias in the camera and assess the plausibility of the mapped debris surface temperature." We did not place the loggers at the surface of the debris layer as their metal housing might have heated up considerably when exposed to direct solar radiation and therefore might record unusually high temperatures that are useless for any kind of comparison.

L150: If you're comparing a ground point measurement to a single thermal pixel at this location and the thermal pixel has a resolution of centimeters, it might be good to take a mean surface temp around the location or move the ribbon off a set distance since it will have a different emissivity to the surrounding rock.

That is true. We did this. However, the procedure is described in a later section (3.6.5) of the manuscript. We therefore included a reference to the respective section.

L151: How long did you leave the sensors running before the survey? At the surface it should be pretty quick, but the sensors need to become isothermal with their environment.

We added this information: "The loggers were set to record temperature at an interval of 10 minutes and were running for about an hour before the survey so that they could become isothermal with their environment."

L159: A few km in the mountains can make a lot of difference, e.g. RH in Table 2. Your final calculations might not be so different with a local sensor, but I think it would be following best practices for a small scale proof of concept study to at least collect local air temperature and RH measurements.

We fully agree that on-site measurements would be beneficial and best practice. Unfortunately, we neither had access to nor funding for the installation of an automatic weather station on the glacier. The experiment was conducted within a student's project without any additional funding.

FIg 4. Should "Ice-snow mask" also include debris (L170)? Was this a manual or automated step?

Thanks for the hint. We updated Fig. 4: "Ice/snow/debris mask"

Fig 4. It's a little confusing if the SfM-MVS steps are open-source or not? L200-214 makes it sound like you used proprietary software but found open source alternatives but didn't find them suitable for your study? I can completely understand how these decisions happen and end up a bit confusing in text form. However, if there is a simple notebook workflow from start to finish, as Fig. 4 suggests, it would be most clear if this was the workflow used, and then you could have a standalone validation section with comparisons to common proprietary software packages.

The inconsistency originates from the history of the workflow development. We failed at the beginning to produce accurate and suitable thermal orthophotos with the open-source pipeline. Anyway, we fully agree and revised the manuscript accordingly. We now present the complete open-source pipeline and have a stand alone validation section using the results from the processing with the proprietary software package.

L198: I don't actually see this step in Fig 4. I guess you are referencing only the open-source pipeline but it reads like the validation step is there.

Yes, we are only referencing the open-source pipeline. We revised the sentence to make that clear.

L198: Already stated in L172

We deleted the sentence.

L240: It's not exactly clear to me where the coefficients came from, from the FLIR algorithm or from a different study or from a post processing step, in which case there should be many for each raw image or only one set from the orthoimage?

We added the information: "An overview of the different atmospheric attenuation constants (provided by FLIR within the metadata) is given in Table..."

Equation 2, 3: can you give the citation again for these? Is it all Tattersall (2021a)?

Yes, it is all Tattersall (2021a). We added the citation to Eqn. 2 and 3.

Equation 4: I don't understand why you take the square of the correction factor?

The reason for the square of the correction factor is that two equations have been combined here. We make it clear in the text.

L255: I'm not so sure if the GCPs meet the criteria, the motivation for the crinkle is to capture surfaces normal to all different angles, in this case I think you're getting almost exclusively a signal from the upper atmosphere or overhead clouds which will be quite a bit colder than reflective temperature from, say, a valley wall or nearby moraine. Especially if only extracting a central point. I think this is also somewhat of a localized measurement problem, e.g. if someone is having a bonfire just off frame, that heat will "bleed" into neighboring pixels. Do you know what, at your scale, the source of this correction is accounting for?

We revised the section: "The thermal GCPs served as a simple diffuse reflector. Hence, the

temperature of each thermal GCP was extracted from the thermal orthophoto calculated as outlined in this section but with emissivity set to 1." It is very difficult to assess the source of the reflected signal detected at the GCPs. Theoretically, longwave radiation reflected by the south-face of the Blüemlisalp, located to the north of the debris-covered area, should be partly reflected into the direction of the camera, but we agree that most of the detected signal probably comes from the cirrus clouds above. Anyway, compared to other uncertainties in the method, the bias related to the reflected apparent temperature is negligible. Varying the reflected apparent temperature of -6.8 °C by ±3.2 °C (standard deviation) would change the surface temperature by only ±0.15 °C.

Table 3: Why do you differentiate between snow and ice if you apply the same emissivity?

Because the northern part of the study area includes an avalanche cone which was covered by fresh snow from a couple of days before the survey (see Fig. 8).

L282: Was there snow in your study area? Last year's or fresh?

See comment before.

L302-308: I think it's fine to have the context here but could move to results or discussion.

Moved to the discussion.

L310: "can be", but within some well known limitations. Time of day, e.g. if all surfaces go isothermal at night, or signal decoupling from thick debris.

We revised the sentence as follows: "A supraglacial debris thickness map can be derived from a corrected thermal orthophoto for areas with a debris thickness of less than ∼0.5 m using either a local empirical relationship between debris thickness and surface temperature or an inverted glacier surface energy balance model, provided that the thermal imagery were collected at a time when the debris surface temperature was sufficiently heterogeneous (e.g. Mihalcea et al., 2008a, b; Foster et al., 2012; Juen et al., 2014; Rounce and McKinney,340 2014; Groos et al., 2017; Rounce et al., 2018, 2021).

L316: Add citation to statement.

We added Mihalcea et al. (2008a).

Figure 5: Add n= to each class; maybe stylistic, but I would continue the curve through the origin since 0 h_d will theoretically pass 0C.

The number of training and validation measurements was included.

We do not continue the curve through the origin as an empirical model is not valid beyond the observations. Moreover, the empirical model would also work for surface temperature measurements with a warm or cold bias and therefore 0 hd does not necessarily passes 0°C. This is only true for a physical model.

L327: Probably best stated with results, putting them into context.

The calculated uncertainties were shifted to the Results.

L355: Thermal diffusivity is at least straightforward to solve for from field data and common in debris cover research. I thought the idea behind your page worth of equations is that there isn't a tuning parameter? Maybe say here that you limited k to realistic values. Unless I'm misreading the purpose of Table 5, add the range/set of values of k rather than just one. Is 1 the best performing that you tried? What did Evatt et al., 2015 use? What metric did you optimize on?

If the debris layer is thin, it is rather difficult to accurately determine the thermal diffusivity as the depth between the loggers is small and measurement uncertainties therefore have a larger impact. However, in a follow-up study, we aim to place loggers at multiple depths in the area where the debris is thickest to get an idea of the thermal diffusivity and conductivity.

Since several of the input parameters were estimated, we had to calibrate the model. We updated this section and included a sensitivity analysis to address the uncertainty related to different input parameters concerning the meteorological conditions and debris properties (see NEW Fig. 13). As the effective thermal conductivity is the parameter that has the largest impact on thin debris in the inverse surface energy balance model (<15 cm), we used it as a tuning parameter. We limited the range of k to realistic values (0.5-1.5 W m$^{-1}$ K$^{-1}$) and calibrated it against the observed debris thickness measurements. In Evatt et al. (2015), k equals 0.585 W m$^{-1}$ K$^{-1}$.

Results

L360: Debris thickness measurements are always a little subjective depending on where you measure. Somewhere, maybe study area section, could you say some qualitative observations of your measurements e.g. in debris less than 1 cm there were fines that still covered the surface or single clasts at the measured debris thickness and bare ice visible between clasts. To those familiar with these field measurements we'll know if this has the potential to have a higher melt rate that bare ice (in the case of the fines) or about the same as bare ice (if it's just stones doing a small scale version of the ice pedestal process).

We added a short sentence in the subsection 3.3.1 (In situ measurements – debris thickness): "Only areas were sampled that were completely covered by debris or fine material."

L360: Maybe "in diameter", of course it's clear as it is, but sounds a little funny.

Revised: "The thickness of the supraglacial debris layer measured manually at 43 points on the Kanderfirn (Fig. 1) ranges from less than 1 cm to about 13 cm (Table A1)."

L362: [were] not measured or mapped.

revised

Figure 6: I think "modeled" not "mapped" on x-axis

changed

Figure 6: The problem with Fig. 6 is that we look to measured for what the distribution should be, but actually the other two are probably more "true" because they are more inclusive. I think you

need to add two more: emp and phy with only intersecting points to measured.

As suggested, we added two more violin plots: emp and phy with only intersecting points to measured.

Figure 7: Same comment as above but compounded by being fundamentally different quantities. Still nice that the shape is similar, but are any of your readers skeptical about the physics behind a thermal image? I'm not 100% sure I see the point of this figure. I think there is no good reason to leave the GCP points in the fig.

As this is a proof-of-concept study, we think it is reasonable to provide a plot that shows the distribution of the mapped surface temperature values. We assume the readers do not question the physics behind a thermal image, but they might be sceptical about the absolute accuracy. The distribution peak close to 0° shows that the measurements are plausible and that there is no major bias.

Figure 8: This figure seems mostly unnecessary and could be merged with Fig. 1. Maybe others feel differently, but I don't get any new information from seeing a raw DSM, the glacier outline is a pretty unrecognizable shape, and I think most specialists reading the paper will take you at your word about high resolution data resolving boulders, debris and ice cliffs/crevasses.

We partly understand your point, but we think this figure helps to better understand the general setting and spatial debris surface temperature and thickness variations. As the other reviewer was wondering about the slope and aspect of the surveyed area, we added two additional maps.

L376-385: Move to Study Area section

We again partly understand your point, but we prefer to keep the section here (same argument as in the previous comment).

L384: give fig ref with labeled "M"

Fig. 8 is now referenced

L384: I have never heard of supraglacial moraines having a lower tail. I think what you mean is a medial moraine structure is crosscut (maybe add: and dispersed) by surface water flowing orthogonal to ice flow. Semantically, all of the rock in your study area is supraglacial moraine.

Thanks, that is exactly what we wanted to say. We adopted your sentence: "In the south-eastern corner of the surveyed area, the lower part of a medial moraine (labelled with "M" in Fig. 8) that is crosscut and dispersed by surface water flowing orthogonal to ice flow and that has become wider over the last years can be seen (Fig. 8; for a better overview see Fig. 11 in Groos et al., 2019)."

L389-390: I don't recall this being a research question raised in the introduction, and I'm not sure if it needs much attention. I would say a network of field based Ts measurements to evaluate the signal of a medium resolution satellite thermal pixel addresses a scientific need, but here I think I trust your drone images to measure Ts more than contact thermistors below the surface. I would

flip the axes of Fig 9 to say you can use T below the rock to approximate Ts :)

As stated before, the uncertainty of the surface measurements as provided by the manufacturer is ±5 °C or 5 % of the reading. We simply conducted this cross-comparison to be sure that the thermal measurements are plausible and that the camera does not exhibit a distinct cold or warm bias.

To the best of our knowledge, the independent (explaining) variable is plotted on the x-axis and the dependent variable on the y-axis. So if we use "T below the rock [debris temperature]" to approximate "Ts [surface temperature]", there is no need to flip the axes.

L399: "More interesting" but in a supplemental figure :) (I'm not saying change it)

"More interesting" with respect to the small-scale deviations in the same supplemental figure, not "more interesting" than the surface temperature map. We referenced Fig. B1 earlier to make that clear.

L403: It's important and good that you used both proprietary and open source software, but slightly concerning that you preference the proprietary while open source is a main contribution of this work.

Agreed! Please refer to our response further above. We revised the manuscript and now present all results from the open-source pipeline while the maps obtained with the proprietary software are only used for comparison.

Figure 10: I thought emissivity was segmented for the different surfaces?

No, it was not segmented in the original version. We now created a surface type raster with the classes ice/snow and debris, and assigned the respective emissivities to create one consistent surface temperature map.

Figure 10: Is there a mechanism for sub freezing temperatures or is that a processing error?

We assume that all snow and ice surface were close to the melting point. So sub-freezing temperatures are likely a signal error.

L404: Notably the snow patch is lower than 0C and what looks like depressions near the bare ice. Any idea if this is signal or error? Maybe places to suggest putting T ground control for future studies?

The lowest surface temperatures were found on the snow patch. However, one should keep in mind that the number of available (thermal) images at the edge of the orthophoto is always reduced. Measurements along the margin are therefore generally less reliable.

L405: I think more technically it is atmospheric temperature not surface temperature. The actual surface temperature of the aluminum is near isothermal with the ground.

The comment is obsolete as we removed the GCPs from the surface temperature map as suggested by the other reviewer and interpolated the area using values from the surrounding

pixels.

L408: What is reporting the average temperature of a random glacier patch for a random time good for? What is a reader learning?

We deleted the sentence.

Figure 11: Do you think it's a coincidence that these very cold areas are on the edges of the spatial domain? Maybe a surface resampling residual? Although that seems unlikely with such dense data. Camera angle / vignetting? I think that might be a problem with smaller thermal cameras vs bigger handheld ones. Can you look at the actual raw thermal tiles and see if they also have that cold signal? What's tripping me up is the surface stream that seems to go cold warm cold, but there is some cross cutting. Could you do an oblique 3D of this so we can see the gradients? The forcings on the ground should be pretty straight forward, 0C water and protection from incoming solar. The change in color of the rock in Fig. 8 at the bottom makes me think it could be a rock wetness / emissivity difference, not so clear about higher/northern cold spots.

We copy the response to the other reviewer here: "We (re)calculated the deviation from the melting point now only for the bare-ice and snow-covered area using the new ice/snow mask (see Fig. 11). We also computed the frequency distribution for the surface temperatures mapped across the bare-ice and snow-covered area (see NEW Fig. 6). The largest deviations of up to ±4 °C can be indeed observed at the edges. This is not surpring as the number of images available for the processing is smaller here and vignetting effects might therefore be more pronounced. In the central part, non-uniformities within individual images associated with external effects on the camera (ambient air, radiation etc.), are probably averaged out during the photogrammetric processing. The deviation patterns shown in Fig. 11 most likely do not represent real surface temperature variations, but rather originate from an insufficient calibration of non-uniformities in the raw thermal images. We elaborate in more detail on the challenges related to UAV-based thermal imaging in the revised discussion, summarise important operational recommendations and outline possible technical solutions. For example, we highly recommend that future studies relying on accurate absolute surface temperatures deploy a portable and light-weight calibrator (heated shutter) that has recently become available and can increase the accuracy of uncooled microbolometers considerably (see Virtue et al. 2021). For an uncooled microbolometer as the one used in this study and taking into account the complexity of thermal imaging on an alpine glacier, the accuracy seems reasonable. Most of the pixel values in the bare ice and snow-covered area (70 %) are in the range of ±2 °C. Only 7 % of the pixels deviate by more than ±3 °C. We added a section in the discussion to elaborate in more detail on the uncertainties in the modelled debris thickness caused by the inaccuracies in the surface temperature map."

L419: That's fine to mention the camera detection limit, but there's a difference between random error and a systematic shift. For your empirical relation this is not really relevant but I think the absolute Ts is important for the EB model.

Yes, accurate absolute Ts is more important for the SEB model than for the empirical model.

L438: Earlier it was not called a sensitivity experiment, and here it's unclear half way through the

| |
|---|
| sentence. There's a difference between a sensitivity analysis and model calibration. |
| That is true. We included a sensitivity analysis in the revised manuscript (see NEW Fig. 13). |
| L438: Consider subsections between empirical and EB results, it's not clear going straight into k that this is a build for EB results. |
| We did not include additional subsections, but we made the transition from the empirical to the SEB results clearer. |
| L440: Point to where in FIg 13 we learn the "very sensitive" |
| See NEW Fig. 13 |
| L440: What does "spread" mean? The RMSE? |
| "spread" was replaced by "RMSE |
| L451: Nice result. Maybe also speak to repeatability / calibration here if there isn't a more in depth section in the discussion. |
| We come back to the difference between the empirical and SEB model in the discussion. |
| Discussion
L454: I'm not recalling any explicit mention of the cost? |
| We included the cost in the section "3.1 Customised low-cost UAV". The deployed DJI Mavic pro costs less than 1,000 EUR, similar to the self-built UAV we use for the annual surveys on the Kanderfirn (see Groos et al. 2019). |
| L459: Trading for an easily acquired time-series. |
| Revised: "However, acquiring a continuous time-series as with automatic field-based thermal imaging techniques is difficult. Moreover, other challenges arise from the deployment of UAVs and both terrestrial and aerial thermal images require a proper calibration and validation." |
| L469: I wouldn't say "defacto infeasible", just a problem needing an innovative solution. |
| Revised |
| L475: Thermal equilibrium with what? Do you mean a single temperature throughout? Your method itself depends on a thermal gradient within the debris layer. |
| No, that is not what we meant. We revised the sentence: "The two subsequent thermal UAV surveys on the Kanderfirn were performed in the early afternoon (between 13:51 and 14:40 CEST) as a high spatial heterogeneity in surface temperatures that reflects supraglacial debris thickness variations can be expected around this time of the day (Bisset et al., 2022)." |
| L530: Why unclear? Within constrained confidence limits, the measurement is not much different from any other that is generally collected without redundancy. |

Revised: "In the absence of independent debris (surface) temperature measurements, such as in the study by Gök et al. (2022), possible biases or shifts in the UAV-based debris surface temperature recordings might be overlooked".

L539-548: This paragraph seems to be repeating established ideas and results, consider cutting.

Minimally shortened

L548: This study is limited to debris thicknesses that are known to be detectable with thermal imagery. I think the key questions for a promising debris thickness mapping technique are centered around thicker debris, so this statement seems too strong to me given the scope of the study. As Ts climbs the upper limb of the Eq in Fig 5, small variations in Ts produce bigger errors in debris thickness. This study, at least the empirical portion, is conveniently distant from the approaching asymptotic behavior.

See general comment at the beginning.

Revised sentence: "As the pronounced spatial surface temperature variations are associated with debris thickness variations, the created thermal orthophoto is a promising basis for high-resolution mapping of supraglacial debris thinner than ∼50 cm, which usually accounts for large parts of the total debris-covered area (Rounce et al., 2021; McCarthy et al., 2022)

L555: Why, because it has a slightly more similar shape than the wider emp? Per my earlier comment, I don't think these plots are directly comparable.

Because the debris thickness map of the pyhsical model better represents the very thin debris thicknesses (ca. 0-3 cm) in the debris-covered area in the south-eastern part of the surveyed area, close to the bare ice surface. Moreover, it also simulates debris thicknesses of more than 15 cm, which can be confirmed from our field visit.

L591: To be fair, Kanderfirn is a tiny glacier with almost no debris cover. That the global dataset caught it at all seems like a very positive review.

From this perspective, yes. We therefore revised the section: "A simple comparison of the high-resolution map and additional field observations from the Kanderfirn with the global debris thickness dataset of Rounce et al. (2021) shows that the global debris mask comprises a quarter of the debris-covered area of the Kanderfirn, although it is relatively small and therefore difficult to detect. However, the modelled debris thicknesses derived from satellite data seem to overestimate considerably the true thickness."

L595-597: Now talking about sub-debris melt seems a little out of place from the bulk discussion of the paper.

We revised this section. It now mainly focuses on debris thickness mapping and less on sub-debris ice melt rates.

L597: I think your study domain would cover 6 ASTER thermal pixels. It's not the most encouraging advice to say this study would have to be repeated maybe 10 times to cover a more traditionally extensive debris cover and evaluate satellite data.

The area surveyed within this proof-of-concept study is of course far to small for any meaningful evaluation of thermal satellite data. However, with a fixed-wing or tail-sitter UAV, debris-covered areas more than 10 times larger as the one presented here can be surveyed and mapped, and would facilitate a comparison with thermal satellite data. We have successfully installed the radiometric TIR camera on our customised low-cost UAV (see Groos et al. 2019) in the meantime and were able to collect radiometric data of a much larger area.

Conclusions

L603: I don't really see any "[paving] the way for glacier-wide high-resolution debris thickness mapping" in this study. The following sentence provides some ideas to overcome the 10 minute drone flight limit, but these are not explored here. The discussion is dismissive of using reanalysis data (Gok et al., 2022) but doesn't put forth an alternative scaling method other than deploying more met stations on glaciers.

We noticed that the scalability and repeatability of the method should be discussed in more detail and, thus, included a specific section on this topic in the discussion. We agree that the complete mapping of large debris-covered outlet glaciers in Alaska or the extensive debris-covered tongues of some glaciers in the Himalaya, Karakoram, Andes etc. is not realistic with the presented methodology, at least not in the near future. We therefore substituted "glacier-wide" by "comprehensive" debris thickness mapping. Because of the following reasons, we are convinced that the presented low-cost and open-source approach paves the way for comprehensive, accurate and high-resolution UAV-based debris thickness mapping:

1. Thermal imaging of larger debris-covered areas is possible with fixed-wing UAVs. We managed in the meantime to install the radiometric TIR camera on our customised low-cost UAV and conduct more extensive surveys. We started to process the data and will present the results in a follow-up paper.

2. Portable and light-weight calibrators that can be attached to a TIR camera to enable consistent thermal measurements and increase the accuracy of absolute temperature recordings are now available (see Virtue et al., 2021) and facilitate more extensive UAV-based thermal imaging.

3. We refrain from using uncorrected global reanalysis data with a resolution of 0.1° x 0.1° for the processing of thermal UAV-data, but not from the use of regional reanalysis data or gridded observational data that better represent the meteorological conditions in the mountains. We also do not opt for more met stations on glaciers. Instead, we suggest that air temperature, relative humidity and radiation could be measured directly by the UAV during thermal imaging to account for spatial and temporal variations in the meteorological/environmental conditions. During a couple of test flights on the glacier in 2022, we managed to acquire thermal images and some meteorological data with a UAV at the same time.

L607: ODM

Modified

L613: I'm not sure I agree this is the most important next step, these thermal data seem quite good and fairly well constrained. I think effort now needs to be focused on thicker debris,

repeatability, and wider coverage.

We agree that follow-up studies should focus on thicker debris, repeatability and wider coverage. However, temporal and spatial variations in the meteorological conditions that affect the accuracy of the surface temperature measurements become more critical for larger and longer aerial surveys. So the technical and practical developments seem to go hand in hand.

L615: Can you state some statistical measures of performance for the two methods here and in the abstract? It would be useful for the reader to see right away how different the end results are of the two methods.

The performance of both methods, the empirical model and calibrated pysical model, is similar. At the location of the in-situ debris thickness measurements, the RMSD of the modelled debris thickness is in the order of 2 cm in both cases. We added this information to the conclusion and abstract.

References

Rounce, David R., et al. "Distributed global debris thickness estimates reveal debris significantly impacts glacier mass balance." Geophysical Research Letters 48.8 (2021): e2020GL091311.

Rounce, David R., et al. "Global glacier change in the 21st century: Every increase in temperature matters." Science 379.6627 (2023): 78-83.

---

## Author Comment (AC2)

**General response**

We thank the editor Kang Yang for obtaining two valuable reviews and Sam Herreid and the anonymous reviewer for their thorough and constructive comments on our manuscript. On the following pages, we address the reviewers' comments point by point. The reviewers' comments are highlighted in grey. We hope that our responses will qualify us to submit a revised version of the manuscript.

**Response to Referee Comment 1 (RC2)**

| |
|---|
| The main strength of the paper is to present an open-source pipeline for the processing of the TIR imagery, which has been an ongoing concern when using the black box, proprietary software to extract temperatures from TIR images. The paper is straightforward and easy to read, and it is well-placed within the existing literature that relates debris-thickness and surface temperature. I enjoyed reading it. |
| Thanks for the positive feedback. |
| I think the manuscript would benefit from a more candid assessment of the performance of their temperature maps, which give results for the snow/ice surface temperatures that seem to have a strong spatially consistent bias, and the applications of both the empirical model and the energy-balance model given the limitations of the input data. Applying these methods is not a straightforward process, which is discussed qualitatively in the paper, but only in general terms. In my opinion, a more quantitative assessment of the model sensitivities would be beneficial. |
| We have taken this comment as an opportunity to perform a simple sensitivity analysis (for the results see NEW Fig. 13), to (re)calculate the surface temperatures for the snow/ice area using a proper mask (see Fig. 11), and to analyse the distribution of the snow/ice surface temperatures (see NEW Fig. 6). |
| Another component that is not much discussed in the paper, but I think should be added, is suggestions on how to upscale this method to a larger domain, considering the limited area tested here, and the possible complications in areas with thicker debris, as the maximum thickness here is 15 cm. |
| We now discuss ideas for the upscaling of this method in more details in the discussion. |
| L9: typo: orthophoto |
| corrected |
| L9: I suggest you mention that you calibrate the energy-balance approach "with an empirical or calibrated inverse surface energy balance". |
| Revised sentence: "Finally, a high-resolution debris thickness map is derived from the corrected thermal orthophoto using an empirical or inverse surface energy balance model that relates |

surface temperature to debris thickness and is calibrated against in-situ measurements."

L84: Could you give the elevation (and elevation range) of the study area, as well as the same characteristics like slope, aspect (which influence the energy-balance application)

We added a sentence with the requested information at the end of the paragraph: "The elevation of the surveyed debris-covered area ranges from 2425 to 2480 m a.s.l. and is cut by two parallel meltwater streams running from northeast to southwest. The inclined areas (mean slope = 15°; standard deviation = 10°) face mainly towards northwest and southeast." Maps of slope and aspect are now also included in Fig. 8.

L117: A strength here is that the flights were so short that it is unlikely that there was a significant change in surface temperature during that time, but it could still have happened. I would like to see somewhere (likely discussion) some mention of possible biases caused by changes in surface temperature during the UAV surveys, especially when it comes to aiming to do longer flights to cover larger areas. This could be made worst in partly overcast weather if cloud movement is occurring rapidly, casting changing shadows, or if flights occur late afternoon or morning. Could you mention if the temperature varied between the flight time (did it warm up or cool down, or was air temperature stable?)

No, according to our debris temperature measurements and the meteorological data from the nearby weather stations, there was no considerable change in air and surface temperature during the short survey period.

We elaborate now in more detail on the potential impact of varying meteorological conditions on the surface temperature measurements in the discussion.

Table 1: Could you change the units from ha to m2 or km2 (as in the text, L119, or at least give the conversion between ha and m2?)

We chose the unit ha for better readability. 1 ha equals 10000 m² or 0.01 km². As ha is a widely used unit for area of land (also in academia) and officially accepted for use with the SI, we think it is not necessary to update Table 1, and also do not mention the conversion factors in the text.

L146: Could you point to the figure comparing measured surface temp to UAV-corrected temperature?

We included a reference to Fig. 9.

L156: few – can you give an actual number?

The sentence was modified: "Two automatic weather station...are located 7 and 5 kilometers away from the Kanderfirn and continuously measure..."

Table 2, and elsewhere: It would be interesting to read a bit more about the uncertainties linked with using such estimates from other locations to derive the energy balance of this highly specific study site. There is a mismatch of complexity here, where you use estimates for the input to the energy-balance model, compared to the high-resolution data you use to derive the debris thickness. I think more information and discussion of the application of the energy-balance model

would be interesting in the results or discussion section because it is not a trivial thing to obtain these results.

We copy and paste here our response to the other reviewer: "We fully agree that on-site measurements would be beneficial and best practice. Unfortunately, we neither had access to nor funding for the installation of an automatic weather station on the glacier. The experiment was conducted within a student's project without any additional funding." We discuss the potential uncertainties regarding the energy balance model in more detail in the Discussion and also provide suggestions for the further improvement of the methodology. In addition, we performed a sensitivity analysis to assess the uncertainties related to different meteorological parameters (i.e. air temperature, incoming short- and longwave radiation, wind speed) and debris properties (i.e. albedo and effective thermal conductivity).

L174: Could you have aimed for lower overlap and achieved a longer flight to cover a larger area? Could this be a suggestion for other studies? (similar to L192)

No, not really. A sufficient overlap is crucial for the generation of accurate orthophotos and DSMs. Instead of reducing the overlap, using fixed-wing or hybrid UAVs that are capable of surveying larger areas are recommended for future studies focusing on debris-thickness mapping.

L209-211: I disagree with this point. The main novel aspect of this paper is presenting an open processing for UAV-based debris thickness but then you don't use the open access too and instead used Pix4D. I think that you should have used only WebODM for the UAV visual UAV instead of using the pix4d if you were only going to use one version. Also, I suggest moving this sentence to the end of the paragraph to avoid talking about thermal, then visual, then thermal again and avoid confusion.

We copy and paste the response to the other reviewer here: "The inconsistency originates from the history of the workflow development. We failed at the beginning to produce accurate and suitable thermal orthophotos with the open-source pipeline. Anyway, we fully agree and revised the manuscript accordingly. We now present the complete open-source pipeline and have a standalone validation section using the results from the processing with the proprietary software package."

L218: according? Do you mean corresponding? Would radiation be radiative?

Yes, we meant corresponding. Modified.

Neither nor. "Radiation temperature" was replaced by "brightness temperature" throughout the text.

L254: This is similar to the approach discussed in Baker et al (2019)?

Baker, EA, Lautz LK, McKenzie JM and Aubry-Wake C (2019) Improving the accuracy of time-lapse thermal infrared imaging for hydrologic applications. Journal of Hydrology 571,60 – 70. doi:10.1016/j.jhydrol.2019.01.053 https://doi.org/10.1016/j.jhydrol.2019.01.053

Thanks for the hint. We included the reference.

Table 4. Any suggestion on how/why they are so variable over such a small area, and for a measurement that occurred all at the same time? What kind of bias occurred if you take the average value for reflected temperature when it is obviously very variable? Was there a spatial pattern to reflect temperature, and could you create a distributed field of reflected apparent temperature?

Good question. We do not have a definite answer, but we could imagine that the main reason for the large spread in the reflected apparent temperature at the different aluminium GCPs is the varying angle, distance and direction between the camera and GCPs during the survey. Most of the GCPs were located slightly off the flight path (see Fig. 1). Although we tried to place the GCPs in flat areas, some of them might have been slightly inclined. Mixed-pixel effects and statistical interpolations during the photogrammetric processing might also be an explanation…

Anyway, compared to other uncertainties in the methodology, the bias related to the reflected apparent temperature is negligible. Varying the reflected apparent temperature of -6.8 °C by ±3.2 °C (standard deviation) would change the surface temperature by only ±0.15 °C.

Figure 5: Could you have a different symbol for the training and validation? It's not the easiest to differentiate them at the moment with the shades of grey.

We updated Fig. 5 and used different symbols for the training and validation data.

L355: You have measured debris thickness and surface temperature from the empirical approach (and near-surface from the in-situ small sensors). Could you calculate keff instead of calibrating it? What kind of values would you get if you tried to derive them from the measurement instead?

As we did not manage to install loggers at different depths in the debris, we cannot calculate keff. We refrain from combining the mapped debris surface temperature and measured nea-surface debris temperatures to calculate keff as the small difference in depth would lead to considerable uncertainties. Moreover, it is unclear how representative k of the 2-cm-thick shale stone would be with respect to keff of the debris layer. Bisset et al. (2022) for example show that keff can vary considerably with depth.

L297: To increase the validity of your approach, you could remove these pixels that you know are not valued by creating a different mask that removed the location of the rocks an

The larger rocks that are scattered across the clean ice are numerous. Digitising and masking them manually would take ages. The only way to detect them automatically would be to apply a temperature threshold, but that's exactly what we did to remove these "outliers" from the statistical analysis.

L361: These scattered boulders – did you remove them from your analysis (removed from your temperature maps) to calculate the debris thickness field? You should probably not use your empirical equation beyond the bounds of the measurements that were used to create your empirical fit, as these modelled thicknesses above 13 cm are extrapolated and not well constrained at all. It looks like you don't have much-modelled thickness above 13cm for the empirical approach, so it might not be a big issue in this case, but something to be careful about.

No, we did not exclude them from the empirical debris thickness map because of practical reasons (see comment above). We agree that results of the empirical model beyond the bounds of the measurements should be interpreted with care, but as Fig. 6 shows, debris thicknesses above 13 cm are almost absent.

Fig 7: I think debris temperature should be before debris thickness in the results, as it is an analysis step that comes before – the measured and mapped temperature influences the modelled thickness, not the other way around. Also, instead, of having outliers in the results that are artifacts of the methods, I think these outliers should be removed from the image by designing a mask that does not include them. This figure presents the debris temperature, so it would be appropriate to remove the GCP from the results.

We agree and exchanged Fig. 6 (NEW Fig. 7) and Fig. 7 (NEW Fig. 6). We created a mask to remove the GCPs from the surface temperature map and interpolated the area of the GCPs using the surrounding pixels and inverse distance weighting (see NEW Section 3.6.4 "Snow and ice masking and GCP correction").

Fig 8. The DSM inset could be called (e) for clarity. In the legend, you set the crevasse and ice cliff as the same feature. Is it the same feature that you refer to, or a different part of the subset image? Can you add what m stands for in the legend/caption?

DSM was called (e). Yes, it is the same feature. The crevasse is the black line and the ice cliffs are the dark greyish areas to the north. M indicates the medial moraine mentioned in the text. The caption was updated accordingly.

L376: How large is it in m x m?

ca. 170 x 390 m

L380: Could you add like Stream 1 and Stream 2 on the fig 8m, like you tag the moraine location?

We labelled Stream 1 and Stream 2 in Fig. 8. and referenced it in the text.

Section 4.4.: It would in good to see if these numbers for the difference in surface temperature are similar to those in other studies that find a bias between TIR and in-situ debris temperature in the discussion section. Is it even a useful way to assess if TIR imagery is correct as they are measuring different things?

In the study by Kraaijenbrink et al. (2018), where measured and mapped debris surface temperatures were compared, the deviation was not quantified (to our knowledge). However, the linear relationship looks similar. We think that temperature loggers installed close to the surface are useful to be sure that the thermal measurements are plausible and that the camera does not exhibit a distinct cold or warm bias. Since they measure different things, the absolute values of the two variables might differ, but they should be highly correlated in any case.

L395: The 162 image number is already mentioned in L194.

| |
|---|
| Deleted |
| L400: A bit contradictory to mention that it is interesting but then put it in the supplementary. |
| "More interesting" with respect to the small-scale deviations in the same supplemental figure, not "more interesting" than the surface temperature map. We referenced Fig. B1 earlier to make that clear. |
| Fig 10-11: I suggest you mask the section of the mages that should not be considered with an emissivity of 0.97. I suggest you segment the cover type (debris, ice) and only show the temperature for the section of the image where the result is valid (where it uses the proper emissivity). only sow the debris are |
| We followed your suggestion and created a surface type raster with the classes ice/snow and debris, and assigned the respective emissivities to create one consistent surface temperature map. In Fig. 11, we now only show the surface temperatures for the surface type classes snow and ice. |
| L403: Similar to the processing of the orthophoto, I think it is misleading to have this paper about an open-access pipeline but then use the result from the proprietary software in the results. I think it would be much more interesting to showcase the result with the open source, and then we could also see the processing artifacts that are mentioned above. |
| We followed your suggestion and revised the manuscript accordingly. |
| L405: I suggest removing this. This is an artifact of the processing that should not be considered a result. |
| Deleted as the GCPs were removed from the surface temperature map (see earlier comment). |
| L408 : given the actual number instead of "about 11" |
| Obsolete as the sentence was deleted. |
| Fig 11, L422-425: I find this result quite concerning. Some patterns make sense: the warmer margin for snow/ice temperature near the debris (a,b,c), but others, such as the snow patch of h-g going from ~+1 to -4 (and likely more if it wasn't masked?), and the strong gradient in temperature between the edge and middle of the image around (c), going from -4 to +4, to -4 over ~150m. To me, these look like edge effects in the processing and suggest that only the middle third of your image is valid. If there is a reason to think that these distributed temperatures are valid, it should be explained. If these are edge effects, then the image should be segmented to only keep the middle section and remove these weird gradients. You mention these artifacts in L422, but then the next sentence states that they perform well enough, and I do not agree with that statement. |
| We (re)calculated the deviation from the melting point now only for the bare-ice and snow-covered area using the new ice/snow mask (see Fig. 11). We also computed the frequency distribution for the surface temperatures mapped across the bare-ice and snow-covered area (see NEW Fig. 6). The largest deviations of up to ±4 °C can be indeed observed at the edges. This is |

not surpring as the number of images available for the processing is smaller here and vignetting effects might therefore be more pronounced. In the central part, non-uniformities within individual images associated with external effects on the camera (ambient air, radiation etc.), are probably averaged out during the photogrammetric processing. The deviation patterns shown in Fig. 11 most likely do not represent real surface temperature variations, but rather originate from an insufficient calibration of non-uniformities in the raw thermal images. We elaborate in more detail on the challenges related to UAV-based thermal imaging in the revised discussion, summarise important operational recommendations and outline possible technical solutions. For example, we highly recommend that future studies relying on accurate absolute surface temperatures deploy a portable and light-weight calibrator (heated shutter) that has recently become available and can increase the accuracy of uncooled microbolometers considerably (see Virtue et al. 2021). For an uncooled microbolometer as the one used in this study and taking into account the complexity of thermal imaging on an alpine glacier, the accuracy seems reasonable. Most of the pixel values in the bare ice and snow-covered area (70 %) are in the range of ±2 °C. Only 7 % of the pixels deviate by more than ±3 °C. We added a section in the discussion to elaborate in more detail on the uncertainties in the modelled debris thickness caused by the inaccuracies in the surface temperature map.

L422-425: Could the relatively good average (0.4C, 0.3C) be linked to the fact that the errors are centred on 0 and so it gives a good average, when in fact it is quite spread out? Could you add information on how you define the +/- for the uncertainty of your numbers? Could you show the distribution of the ice and snow temperature in Figure 7, in addition to the map of the whole area (maybe even show both the snow/ice segmented distribution and the debris mask-only distribution?)

We added the different distributions (snow/ice, debris, total area) in the NEW Fig. 6 (previous 7). See also comment above. We used the standard deviation of the individual pixel values (mapped surface temperature minus melting point) as a measure for uncertainty. The average/median slightly above the melting point indicates that there is no consistent bias or shift in the temperature measurements. In our view, the standard devidation in the order of ±1 °C is reasonbale for an uncooled microbolometer and first attempt, and justifies the use of the mapped surface temperatures for debris thickness modelling. For comparison, Gök et al. (2023), who performed thermal imaging on a different glacier in Switzerland, state a mean ice surface temperature of 0.72-2.26 °C (and standard deviation of up to 2.87 °C).

L422-425: My understanding and experience is also that thermal infrared cameras can be quite accurate from pixel to pixel within one image, but can be quite off in terms of absolute temperature. It makes me more cautious about these spatial patterns in the temperature of the ice surface. I understand that the camera accuracy says +5 to -5, but that covers a much too large range and really limits the possibility to investigate TIR use for glacier melt, where a much smaller temperature range has large consequences for melt.

The camera accuracy given by the manufacturer is ±5 °C or 5% of the reading in the range of -25 to +135 °C. Our results indicate that the accuracy in the range of -5 to +35 °C seem to be much better (probably ±1-2 °C), depending of course on the ambient conditions during the survey.

| |
|---|
|  |

Fig 11: Also, maybe put a dotted box around the area that is used for the quantitative assessment – is that the area near where the temperature artifact is in (g)?

We created an ice/snow mask and perform the quantitative assessment only for this area (see NEW Fig. 6 and Fig. 11).

L429: couple millimetres -> use the actual number?

Updated: "... ranges from around 1 centimeter up to 15.5 cm (Fig. 6)."

L431: relatively thick -> How thick?

Updated: "Besides this, the debris layer appears to be relatively thick (ca. 5-10~cm) in the elevated area between the parallel supraglacial meltwater streams (Fig. 12d)."

L435: Could they be linked to wetness level, which would influence the conductivity and the surface temperature, instead of thickness?

We cannot completely rule out this alternative explanation, but we think it is rather unlikely. The visual orthophoto does not indicate any spatial variations in the wettnes level. Moreover, if the wettnes level would vary in space, then the question would be wich condition could lead to the observed stripe-like pattern? Besides thickness, also the grain size, porosity and lithology of the debris layer could alter the water level/content. But then again the question would be which process leads to stripe-like variations in grain size, porosity, lithology etc.

L440: Have you tested how other parameters are sensitive in the model? Can you justify calibrating this one instead of a selection of other parameters?

Yes, we also tested other parameters (air temperature, shortwave radiation, longwave radiation, wind speed and debris albedo) in the surface energy balance model (see NEW Fig. 13). The model responds sensitively to the tested parameters, but the uncertainty introduced by these parameters concerns mainly thick debris (>10 cm). Since the (effective) thermal conductivity is the parameter that is most critical for thin debris (<10 cm), which is characteristic for the studied glacier, we chose this one for the model calibration.

L440: I find it interesting that you use a fairly complex inverse energy-balance approach, but then calibrate it with one parameter to fit the data. The other component of that model is also highly uncertain – meteorology, albedo, etc, are all very specific to the study area, and potentially even variable throughout your study area. It would be interesting to hear more about how suitable it is to use spatially homogenous input to the model when you are looking at a variable terrain. You mention this very briefly in L445, but maybe you could elaborate slightly more in the discussion.

We used the theoretical model of Evatt et al. (2015) as it is the only one that is able to reproduce the characteristic features of the empirical Østrem curve. Although it is not the main point of this comment, we would like to emphasise that the complex model used here could be easily replaced by any other model in the presented open-source pipeline.

While some of the meteorological data (e.g. air temperature or incoming shortwave radiation) extrapolated from the weather stations nearby probably describe the local conditions on the

glacier fairly well, other parameters (such as incoming longwave radiation or wind speed) are subject to higher uncertainties. It is also true that some of the input parameters (for example debris albedo or air tmperature) can be expected to vary across the surveyed area. We elobrate more on the related uncertainties and the refinement of the methodology.

L470: But, if you have a fixed wing that can flight longer and further, you are likely to be able to find a patch or snow or smooth meadow where you can land adjacent to the glacier…. Another limitation to uav is that they are realy bulking to hike in to remote sites.

Theoretically yes, but

1) the legal framework in almost all countries does not support flying out of sight (or at least requires a comprehensive safety concept)

2) flying out of sight in mountainous terrain is risky

3) smooth areas are often difficult to find in glacierised and deeply-incised valleys

Yes, they are bulky, but considering the constant technological advancements in the field, smaller high-endurance UAVs that are more suitable for applications in the mountains should become available in the upcoming years...

Figure 15: I don't think this figure is needed as you have the RMSE values on fig 13.

It is not necessarily needed, but as it is small and nicely shows the model improvement and the remaining error related to other uncertainties in the workflow, we prefer to keep it in the article.

L476: Aubry-Wake et al., 2022 might be a helpful reference about the different factors that influence TIR acquisition for debris thickness measurements because the conclusions are very different – midday is not a good time to have a strong relationship between surface temperature and debris thickness, but you focus on very thin debris overall, so a different dataset!

Aubry-Wake, C., Lamontagne-Hallé, P., Baraër, M., McKenzie, J., & Pomeroy, J. (2023). Using ground-based thermal imagery to estimate debris thickness over glacial ice: Fieldwork considerations to improve the effectiveness. Journal of Glaciology, 69(274), 353-369. doi:10.1017/jog.2022.67

Thanks for pointing out this publication. We were not aware of it before and have considered it now in the discussion.

L486: A nuance that I think needs to be clarified here is that precise surface temperatures (that are consistent together) are needed for empirical thickness calculation, but for empirical calculation, the measurements do not need to be accurate. They could be biased, but as long as they are consistent, it works. However, for energy-balance approaches, you need both accurate and precise measurements of surface temperature.

Yes, accurate surface temperature are less important for the empirical approach than for the physical approach, but temporal changes in the environmental and meteorological conditions would nevertheless also affect the empirical relationship between Ts and hd. We therefore slightly adjusted the sentence: "Since accurate and consistent surface temperatures are..."

L518: But those snow/ice temperature maps still show a large vignetting effect, so potentially these atmospheric corrections and reflected radiation are not enough to obtain precise and consistent temperature maps. These vignette effects have not been mentioned in ground-based measurements, so they might originate from the processing of the UAV images.

Vignetting effects might be more pronounced in UAV-based than in ground-based infrared thermography as the camera is more exposed to changes in the ambient conditions during the flight than at one (maybe wind-shielded) location on the ground. The larger surface temperature deviations along the margin of the orthophoto might be related to vignetting effects or in general to changes in ambient conditions (e.g. wind speed, differential heating of the camera housing related to flight direct and position of the sun). The effect is probably less pronounced in the central part of the orthophoto as the non-uniformities in individual photos should be averaged out to a certain degree during the photogrammetric processing in areas where sufficient thermal images are available. However, overall the accuracy seems reasonable for an uncooled microbolometer. The middle boxplot in NEW Fig. 6 indicates that 50 % of the surface temperature values of the ice and snow area are in the range from -0.9 to 1.8 °C. Nevertheless, we recommend that future studies that rely on accurate surface temperature measurements deploy a portable and light-weight calibrator as presented in Virtue et al. (2021) to get rid of non-uniformities in the individual images.

L530: This might be a good moment to point out that using small in-situ temperature sensors tucked in the debris, as you did, has a fairly limited use to assess the performance of UAV-based temperature.

As stated before, we think that installing temperature loggers close to the surface is justified in proof-of-concept study to be sure that the thermal measurements are plausible and that the camera does not exhibit a distinct cold or warm bias. However, we agree that the loggers are dispensable once the robustness and reliability of the camera in harsh environments has been confirmed by several studies.

We revised the sentence: "In the absence of independent debris (surface) temperature measurements, such as in the study by Gök et al. (2022), possible biases or shifts in the UAV-based debris surface temperature recordings might be overlooked".

L558: But even applying a site-specific empirical model can lead to erroneous debris thickness if the model is based on flawed data, like a bias in the sampling of the debris thickness, or a surface temperature that does not correlate well with debris thickness due to time of day, as discussed by Herreid (2022) and Aubry-Wake et al. (2023).

Yes, the overall methodology depends in principle on a strong correlation of surface temperature and debris thickness. We discuss this aspect now in more detail.

L559: But you do not account for spatial variation right? I appreciate that you discuss how it would be nicer to have in-situ measurements of meteorology, I would like a bit more quantitative analysis of this. For example, what kind of error or uncertainty are you erasing by calibrating keff? Is the model as sensitive to air temperature as it is keff? Applying an energy balance model to a small area like this, with high spatial heterogeneity, is hard to get right even with data coming

from the site, so it would be good to have a bit more information on the sensitivity of the debris thickness to the model application.

No, we have not accounted for spatial variations in this study. However, the usage of gridded data instead of single parameter values can be easily integrated in the presented open-source pipeline.

We managed to measure spatial air temperature variations in parallel with the thermal imaging in a follow-up experiment, but haven't completed the analysis yet. Please refer to the new sections in the revised manuscript and to the comments further above regarding the model sensitivity.